# The protein arginine methyltransferase PRMT9 attenuates MAVS activation through arginine methylation

Xuemei Bai[1], Chao Sui[1], Feng Liu[1], Tian Chen[2], Lei Zhang[1], Yi Zheng[1], Bingyu Liu[1] ✉ & Chengjiang Gao [1] ✉

The signaling adaptor MAVS forms prion-like aggregates to activate the innate antiviral immune response after viral infection. However, spontaneous aggregation of MAVS can lead to autoimmune diseases. The molecular mechanism that prevents MAVS from spontaneous aggregation in resting cells has been enigmatic. Here we report that protein arginine methyltransferase 9 targets MAVS directly and catalyzes the arginine methylation of MAVS at the Arg41 and Arg43. In the resting state, this modification inhibits MAVS aggregation and autoactivation of MAVS. Upon virus infection, PRMT9 dissociates from the mitochondria, leading to the aggregation and activation of MAVS. Our study implicates a form of post-translational modification on MAVS, which can keep MAVS inactive in physiological conditions to maintain innate immune homeostasis.

Type-I interferons (IFN) are critical in host defense against invading pathogens in innate immunity, and are especially protective in acute viral infections[1–3]. In response to viral infection, pattern-recognition receptors (PRR) recognition of the pathogen-associated molecular patterns (PAMP) triggers the production of type-I IFNs through the distinct downstream signaling pathways. RIG-I-like receptors (RLR) have a vital function in the sensing of cytosolic viral RNA[4,5]. To date, three RLRs members have been identified: RIG-I, MDA5 and LGP2. Upon binding to the cytosolic RNA, RIG-I and MDA5 binds to the mitochondrial antiviral signaling protein (MAVS, also known as IPS-I / VISA/CARDIF) and induce MAVS to form prion-like aggregates[6,7]. Aggregation of MAVS further induces the activation of the TANK-binding kinase 1 (TBK1) and cytosolic kinases IKK, which then activate the transcription factors IRF3 and NF-κB and result in the production of type-I IFNs and pro-inflammatory cytokines to defend against viral proliferation[8,9].

MAVS has a C-terminal transmembrane (TM) domain that anchors to the mitochondrial outer membrane; a middle proline-rich region (PRR) that binds to the tumor necrosis factor receptor-related factor (TRAF) family members; and an N-terminal caspase-recruitment domain (CARD) that interacts with the CARDs of RLRs[6,10]. As indicated by a number of existing studies, MAVS is the key adapter protein of RLRs for defend against RNA viruses[9,11]. Additionally, spontaneous aggregation of MAVS has been reported to be associated with autoimmune diseases. For example, one study reported that spontaneous MAVS oligomerization was observed in the plasma of SLE patients, but rarely in healthy control patients[12]. Moreover, in the same study, the MAVS-C79F variant failed to oligomerize and showed exhibited milder forms of SLE, suggesting that the inhibition of spontaneous MAVS oligomerization might be a therapeutic target in SLE[12,13]. According to a recent report, inappropriate or persistent MAVS aggregation led to increased production of IFN-I and systemic autoimmunity in a significant fraction of SLE patients[14]. Hence, efficient activation of MAVS is crucial for mediating the host innate immunity against viral infection and must be tightly regulated to avoid potentially harmful tissue damage. The posttranslational modifications (PTM) of MAVS have an important function in modulating the activity of MAVS. For instance, phosphorylation of MAVS is dynamically regulated by kinases TBK1 and phosphatase PPM1A during viral infections[9,11] to fine-tune the type I interferon production. With TRIM31-mediated K63-linked

[1]Key Laboratory of Infection and Immunity of Shandong Province & Department of Immunology, School of Basic Medical Sciences, Shandong University, Jinan, Shandong 250012, PR China. [2]Department of Pathogenic Biology, School of Biomedical Sciences, Shandong University, Jinan, Shandong 250012, PR China. ✉e-mail: liubingyu@sdu.edu.cn; cgao@sdu.edu.cn

ubiquitination of MAVS, the aggregation of MAVS is promoted after viral infections[15]. The interaction between TRIM31 and MAVS is positively modulated by O-GlcNAcylation of MAVS[16]. TRIM21 catalyzes the K27-linked polyubiquitination of MAVS, thereby promoting the recruitment of TBK1 to MAVS and enhancing the innate immune response[17]. A couple of deubiquitinases are reported to regulate the activation of MAVS. Ovarian tumor family deubiquitinase 4 (OTUD4) removes K48-linked polyubiquitin from MAVS, while OTUD3 and YOD1 are responsible for cleaving K63-linked polyubiquitin chains from MAVS[18,19]. Different from the above deubiquitinases, USP18 can upregulate the K63-linked polyubiquitination of MAVS to promote the type I interferon response[20]. At present, several types of PTMs including phosphorylation, ubiquitination, and O-GlcNAcylation have been reported to regulate MAVS signaling. More recently, protein arginine methylation, a significant PTM in regulating multiple intracellular signaling, has been reported to regulate IFN signaling[21–24]. Whether MAVS is arginine methylated and which arginine methyltransferase controls its arginine methylation warrant investigation.

In the present study, protein arginine methyltransferase PRMT9 is identified as a negative regulator of innate antiviral immunity. PRMT9 deficiency enhances the innate antiviral response to RNA viruses both in vitro and in vivo. Our data show that PRMT9 can catalyze the arginine methylation of MAVS at Arg41 and Arg43. These results further demonstrate that methylation of MAVS by PRMT9 can inhibit MAVS aggregation in the presence or absence of viral stimulation. Collectively, the present study indicates a mechanism that keeps MAVS inactive under physiological conditions.

## Results

### PRMT9 negatively regulates RLRs-induced IFN-β signaling

Mitochondria are critical in the cellular innate immune response against viral infection[15]. To identify the potential function of PRMTs in innate antiviral immunity, GFP-PRMT1-9 plasmids were transfected together with DsRED2-Mito plasmid into HEK293T cells for 24 h, followed by mock infection or infection with SeV for 8 h, and the mitochondrial localization was measured. Confocal microscopy showed that PRMT1, PRMT2, PRMT3, PRMT4, PRMT5, PRMT6 and PRMT8 failed to co-localize with the mitochondria and the localization was not affected by SeV infection. While PRMT7 colocalized with mitochondria in HEK293T cells and was unaffected by SeV infection. Notably, PRMT9 colocalized with the mitochondria and dissociated from the mitochondria upon SeV infection in HEK293T cells (Supplementary Fig. 1). Further, we also observed that the colocalization of the endogenous PRMT9 with mitochondria was attenuated following the SeV infection in PM or THP-1 cells (Supplementary Fig. 2e, f). Next, crude mitochondria were extracted from peritoneal macrophages (PM) or THP1 cells, and immunoblot analysis of the expression of PRMTs in total cell lysates or crude mitochondria lysates infected with SeV for 0–12 h was performed. The results of the immunoblots analysis (Supplementary Fig. 2a–d) are consistent with the confocal microscope images in PM and THP1 cells, suggesting that PRMT9 localized in mitochondria, and the amount of mitochondrial PRMT9 was gradually decreased upon viral infection. These data suggest that PRMT9 is a mitochondrion-associated protein and may involve in the regulation of anti-RNA virus infection.

To investigate the physiological action of PRMT9 in antiviral signaling, a small interfering RNA (siRNA) that targeted mouse PRMT9 was designed, and siRNA was transfected into peritoneal macrophages following infection with SeV and VSV or stimulation with RNA mimics 5'ppp-RNA. The siRNA knockdown efficiency was validated at both the mRNA level and protein level (Supplementary Fig. 3a). Findings were made that siRNA knockdown of Prmt9 expression in primary macrophages increased the SeV-, VSV- and 5'ppp-RNA-induced mRNA expression levels of Ifnb1, Ifna4, Cxcl10 (Supplementary Fig. 3b–d). Notably, SeV-, VSV- and 5'ppp-RNA-induced IFN-β proteins were also increased in siPRMT9 transfected macrophages compared with that in control siRNA transfected cells (Supplementary Fig. 3b–d). In contrast, the replication of VSV was attenuated in siPRMT9 transfected macrophages compared with that in control siRNA transfected cells (Supplementary Fig. 3e).

siRNA targeting human PRMT9 was also designed to further investigate whether PRMT9 acts in different species, and siPRMT9 was transfected into human THP-1 monocytic cells (Supplementary Fig. 3f). We found that the SeV- and 5'ppp-RNA-induced mRNA expression levels of IFNB1 were also increased in THP-1 cells transfected with siPRMT9 compared with that in the cells transfected with control siRNA (Supplementary Fig. 3g). The VSV-induced mRNA expression levels of IFNB1, IFNA4 and CXCL10 were also increased in siPRMT9 transfected THP-1 cells (Supplementary Fig. 3h). Meanwhile, the replication of VSV in THP-1 cells was attenuated upon knockdown of PRMT9 expression (Supplementary Fig. 3i).

To further confirm the function of PRMT9, two PRMT9 knockout RAW264.7 cell lines were constructed by means of the CRISP/Cas9 technique. The results of western blotting showed that the successful knockout of PRMT9 protein in RAW264.7 cells (Fig. 1a). Consistent with the siRNA knockdown results, the mRNA expression levels of Ifnb1, Ifna4 and Cxcl10 were upregulated in PRMT9 KO RAW264.7 cells after SeV infection or 5'-pppRNA transfection (Fig. 1b, c). Similarly, knockout of Prmt9 in RAW264.7 cells also increased VSV-induced Ifnb1 expression (Fig. 1d). At the same time, the replication of VSV was attenuated in PRMT9 KO RAW264.7 cells (Fig. 1d, e).

Flag-PRMT9 was also transfected in HEK293T cells, and the overexpression of Flag-PRMT9 in HEK293T cells was found to significantly decrease the expression of IFNB1 mRNA upon SeV or VSV infection (Fig. 1f, g). At the same time, VSV replication was increased in HEK293T cells with overexpression of Flag-PRMT9 (Fig. 1g). The results of fluorescence microscopy and flow cytometric analysis showed that overexpression of PRMT9 in HEK293T cells facilitated VSV-GFP replication (Fig. 1h). As previously reported[25], miRNA-543 directly binds to 3'-UTR of PRMT9 mRNA to inhibit PRMT9 translation. Indeed, we found that overexpression of miR-543 significantly decreased PRMT9 expression. Consistently, overexpression of miR-543 significantly increased the Ifnb1, Ifna4, and Cxcl10 mRNA levels after infection with SeV, VSV or stimulation with 5'PPP-RNA compared with the NC group in macrophages (Supplementary Fig. 3j, k). Overall, these data suggest that PRMT9 negatively regulates RLRs-induced IFN-β signaling to promote RNA virus infection.

### PRMT9 deficiency enhances RLRs-triggered IFN-β signaling

To further investigate the function of PRMT9 in the regulation of viral infection, the Prmt9$^{fl/fl}$ mice were crossed with Lyz2-Cre transgenic mice to produce myeloid-specific PRMT9 knockout mice. Primary peritoneal macrophages were prepared from Prmt9$^{fl/fl}$ Lyz2-Cre (hereinafter referred to as 'Prmt9$^{CKO}$') and Prmt9$^{fl/fl}$ mice (hereinafter referred to as 'Prmt9$^{WT}$') followed infection with SeV, VSV, and HSV-1 or stimulation with 5'-pppRNA. Consistent with the observation from the siRNA knockdown or sgRNA knockout of PRMT9, infection of Prmt9$^{CKO}$ peritoneal macrophages with SeV or stimulation with 5'PPP RNA led to a significant increase in the expression of Ifnb1, Ifna4, and Cxcl10 mRNA compared with that in Prmt9$^{WT}$ macrophages (Fig. 2a, b). IFN-β secretion was also upregulated in Prmt9$^{CKO}$ macrophages than that in Prmt9$^{WT}$ macrophages (Fig. 2a, b). However, there were no differences observed in the levels of HSV-1 infection-induced type I IFNs and Cxcl10 expression between Prmt9$^{WT}$ macrophages and Prmt9$^{CKO}$ macrophages (Fig. 2c). The results further showed that the Ifnb1 mRNA level was increased in Prmt9$^{CKO}$ macrophages after infection with VSV, whereas VSV mRNA, VSV titers and VSV protein were significantly decreased in Prmt9$^{CKO}$ macrophages (Fig. 2d, e). Collectively, a conclusion could be drawn that PRMT9 negatively regulated RLRs-induced IFN-β signaling.

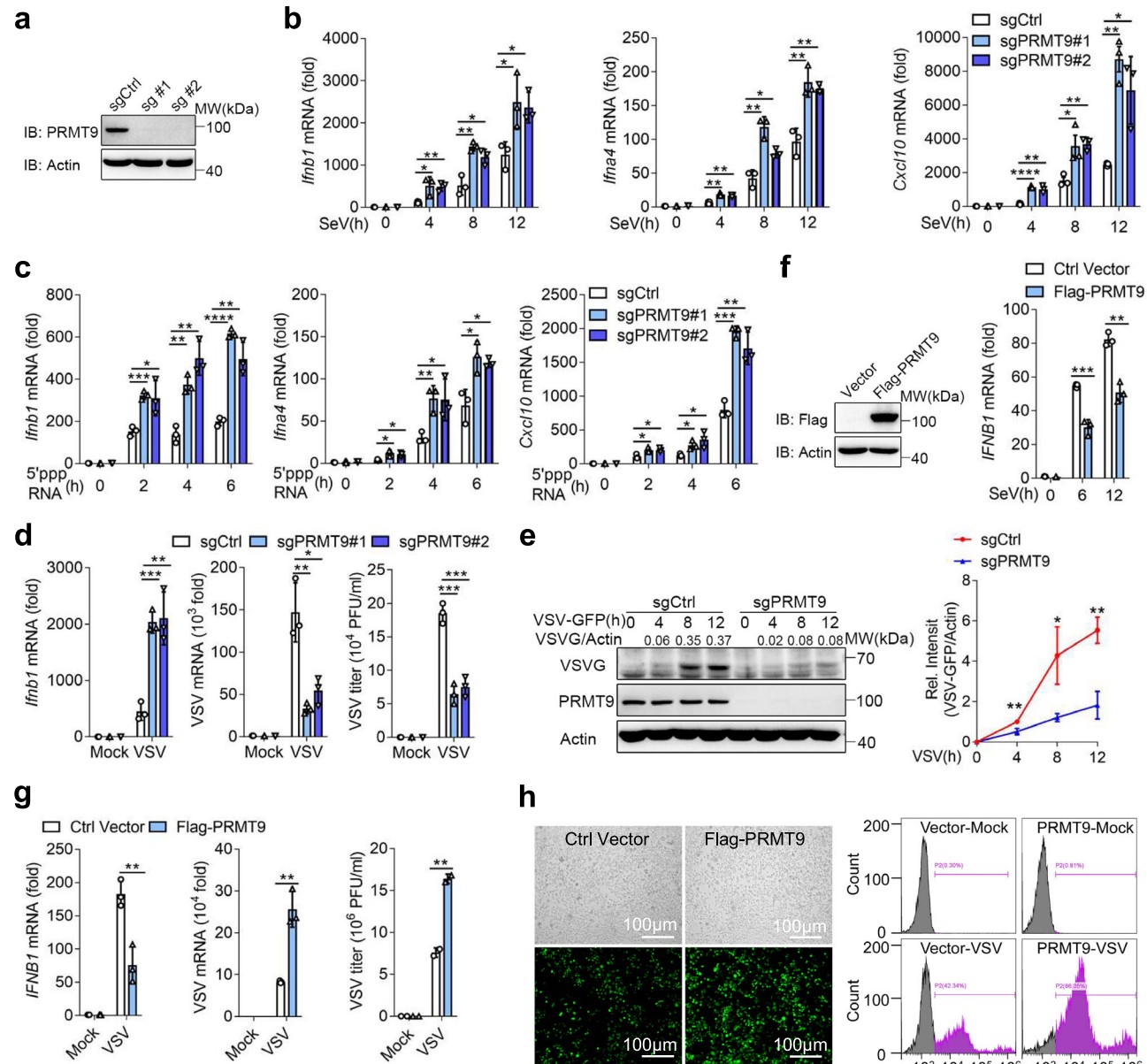

**Fig. 1 | PRMT9 negatively regulates RLRs-induced IFN-β production.**
**a** Immunoblot analysis of PRMT9 in *Prmt9*-knockout RAW264.7 cell lines. **b, c** qRT-
PCR analysis the expression of *Ifnb1*, *Ifna4* or *Cxcl10* mRNA in *Prmt9*-knockout cells
with SeV infection or stimulated with 5′-pppRNA (mean ± SD, two-tailed student's *t*
test was performed, for **b**, *Ifnb1*: *$p$ = 0.0351, **$p$ = 0.0049, **$p$ = 0.0032, *$p$ = 0.0160,
*$p$ = 0.0385, *$p$ = 0.0149 in sequence; *Ifna4*: **$p$ = 0.0020, **$p$ = 0.0029, **$p$ = 0.0026,
*$p$ = 0.0104, **$p$ = 0.0067, **$p$ = 0.0030 in sequence; *Cxcl10*: ****$p$ < 0.0001,
**$p$ = 0.0024, *$p$ = 0.0380, **$p$ = 0.0019; **$p$ = 0.0012, *$p$ = 0.0192 in sequence. For
**c**, *Ifnb1*: ***$p$ = 0.0006, *$p$ = 0.0341, **$p$ = 0.0023, **$p$ = 0.0022, ****$P$ < 0.0001,
**$p$ = 0.0033 in sequence; *Ifna4*: *$p$ = 0.0155, *$p$ = 0.0175, **$p$ = 0.0081, *$p$ = 0.0382,
*$p$ = 0.0199, *$p$ = 0.0117 in sequence; *Cxcl10*: *$P$ = 0.0246, *$p$ = 0.0195, *$p$ = 0.0212,
*$p$ = 0.0332, ***$p$ = 0.0002, **$p$ = 0.0043 in sequence; $n$ = 3 independent experi-
ments). **d** qRT-PCR analysis of *Ifnb1* (left), VSV mRNA (middle) and plaque assay of
VSV titers (right) in *Prmt9*-knockout RAW264.7 cells infected with VSV (MOI (mul-
tiplicity of infection), 0.1), and (**e**) Immunoblot analysis of VSV glycoprotein (VSV-G)
in lysates of RAW264.7 cell lines infected with VSV for 0-12 h. For the densitometric
analysis (right), VSV bands were normalized with individual actin, line graphs were
presented relative to the second lane (mean ± SD, two-tailed student's *t*-test was
performed, for **d**, *Ifnb1*: ***$p$ = 0.0004, **$p$ = 0.0044; VSV mRNA: **$p$ = 0.0052,

*$p$ = 0.0144; VSV titer: ***$p$ = 0.0006, 0.0009 in sequence. The data shown in **e** are
from one representative experiment of at least 3 biological independent experi-
ments, two-tailed student's *t*-test was performed, for **e**, **$p$ = 0.0062, *$p$ = 0.0209,
**$p$ = 0.0024 in sequence). **f** Immunoblot analysis of Flag-PRMT9 (left) in contrl
HEK293T cells and Flag-PRMT9-overexpressing HEK293T cells. qRT-PCR analysis of
*IFNB1* mRNA in HEK293T cells transfected for 24 h with those plasmids, followed by
infection with SeV. **g** qRT-PCR analysis of *IFNB1* mRNA (left) or VSV mRNA (middle),
and plaque assay of VSV titers (right) in HEK293T cells transfected with control
plasmid (Ctrl) or plasmid expressing Flag-PRMT9 for 24 h, followed by infection
with VSV (mean ± SD, two-tailed student's *t*-test was performed, for **f**, ***$p$ = 0.0008,
**$p$ = 0.0016 in sequence; for **g**: **$p$ = 0.0059, 0.0022, 0.0041 in sequence; $n$ = 3
independent experiments). **h** Microscopy imaging (left) and Flow cytometry ana-
lysis (right) of the replication of GFP-VSV in HEK293T cells transfected with control
plasmid (Ctrl) or plasmid expressing Flag-PRMT9 for 24 h, stimulated with VSV-GFP
for 18 h, bars: 100 μm. The qRT-PCR results are presented relative to those of
untreated wild-type cells or transfected with a Vector plasmid (Average of three
replicates, **b**–**d**, **f**, **g**). The data shown in **a**, **e**, **f**, and **h** are from one representative
experiment of at least 3 biological independent experiments. Two-tailed Student's
*t*-test was performed, with *$P$ < 0.05; **$P$ < 0.01; ***$P$ < 0.001; ****$P$ < 0.0001 (**b**–**g**).

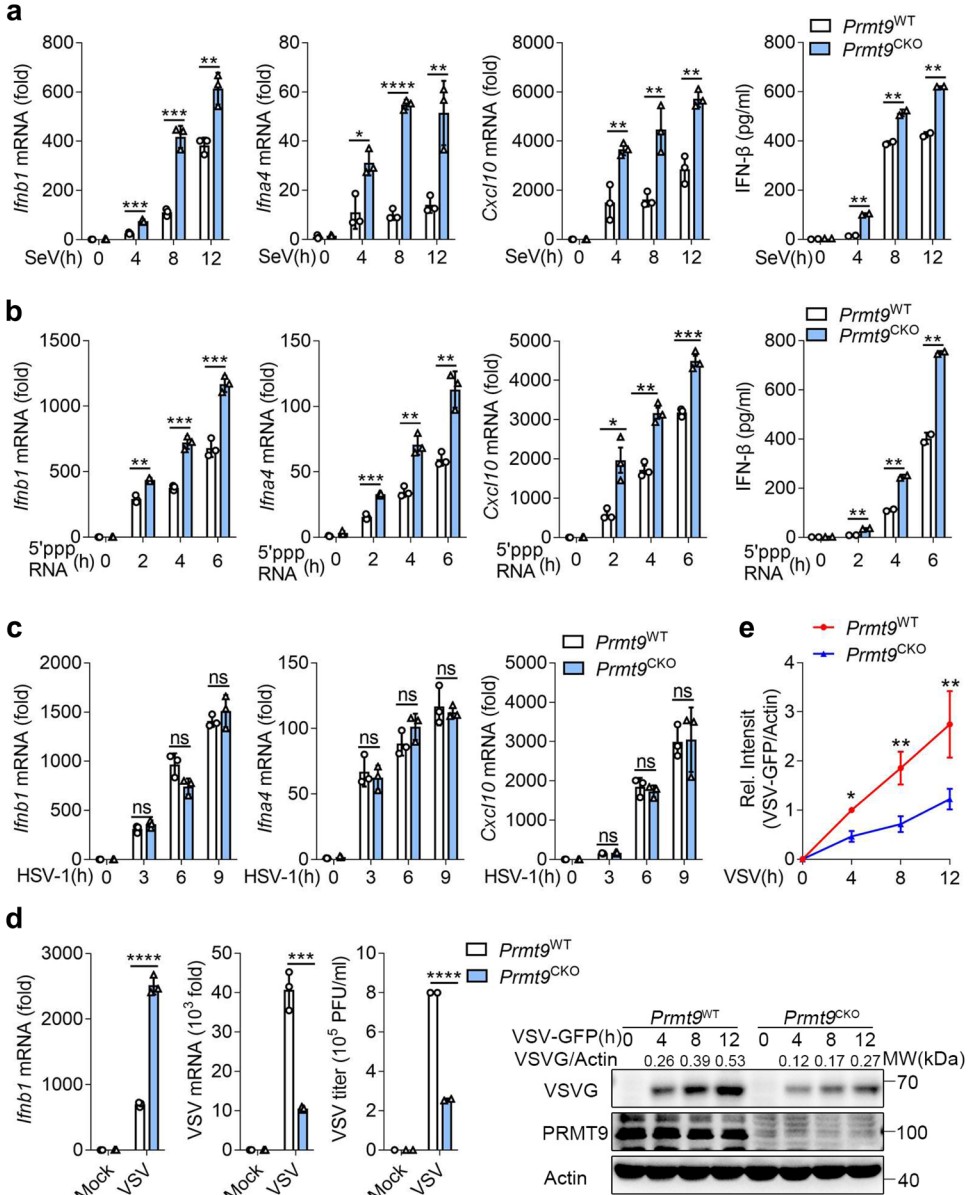

**Fig. 2 | PRMT9 deficiency in primary peritoneal macrophages enhances RLRs-triggered IFN-β signaling. a, b** ELISA quantification of IFN-β secretion and (**a**–**c**) qRT-PCR analysis of *Ifnb1*, *Ifna4*, *Cxcl10* mRNA in *Prmt9*CKO and *Prmt9*WT peritoneal macrophages infected with **a** SeV or **b** stimulated with 5′-pppRNA or infected with **c** HSV-1 (mean ± SD, two-tailed student's *t*-test *Prmt9*CKO vs. *Prmt9*WT, for **a**, *Ifnb1*: ***p = 0.0002, ***p = 0.0004, **p = 0.0052 in sequence; *Ifna4*: *p = 0.0140, ****p < 0.0001, **p = 0.0088 in sequence; *Cxcl10*: **p = 0.0078, 0.0076, 0.0017 in sequence; IFN-β: **p = 0.0012, 0.0081, 0.0015 in sequence, n = 3 independent experiments. For **b**: *Ifnb1*: **p = 0.0011, ***p = 0.0003, ***p = 0.0008 in sequence; *Ifna4*: ***p = 0.0002, **p = 0.0020, **p = 0.0034 in sequence; *Cxcl10*: *p = 0.0139, **p = 0.0013, ***p = 0.0008 in sequence; IFN-β: **p = 0.0034, 0.0024, 0.0024 in sequence, n = 3 independent

experiments). **d** qRT-PCR analysis of *Ifnb1* (left), VSV mRNA (middle), plaque assay of VSV titers (right) and immunoblot analysis of VSV-G (far right) in *Prmt9*CKO and *Prmt9*WT peritoneal macrophages infected with VSV (MOI, 0.1) for 0–12 h (mean ± SD, two-tailed student's *t*-test *Prmt9*CKO vs. *Prmt9*WT, ****p < 0.0001, ***p = 0.0005, ****p < 0.0001 in sequence, n = 3 independent experiments). **e** For the densitometric analysis (right), the values were normalized to actin (mean ± SD, two-tailed *t*-test *Prmt9*CKO vs. *Prmt9*WT, *p = 0.0248, **p = 0.0020, 0.0046 in sequence, n = 3 independent experiments). Line graphs were presented relative to the second lane. The qRT-PCR and ELISA results are presented relative to those of untreated wild-type cells (Average of three replicates, **a**–**d**). *P < 0.05; **P < 0.01; ***P < 0.001; ****P < 0.0001; ns not significant (two-tailed Student's *t*-test).

## PRMT9 inhibits innate antiviral response in vivo

To definitively confirm the physiological function of PRMT9 in RNA virus infection, *Prmt9*CKO mice were infected with VSV by means of tail vein injection. As shown in Fig. 3a, the expression levels of *Ifnb1* in the lung, liver, and spleen of *Prmt9*CKO mice were significantly higher than those of *Prmt9*WT mice organs after infection with VSV. Additionally, ELISA results also showed that the level of IFN-β protein in serum from *Prmt9*CKO mice was significantly increased compared with that from *Prmt9*WT mice 24 h after VSV infection (Fig. 3b). Further, Plaque

assays for VSV titers and qRT-PCR analysis for VSV mRNA confirmed that the VSV replication was significantly attenuated in the lung, liver, and spleen in *Prmt9*CKO mice compared with those in *Prmt9*WT mice (Fig. 3c, d). Moreover, *Prmt9*CKO mice were found to be less susceptible than *Prmt9*WT mice to infection with VSV but not to infection with HSV-1 (Fig. 3e, f). The results of hematoxylin-and-eosin staining showed that there was no significant difference in the lung between *Prmt9*WT and *Prmt9*CKO mice without VSV infection, and PRMT9 deficiency alleviated inflammatory cell infiltration, tissue edema and

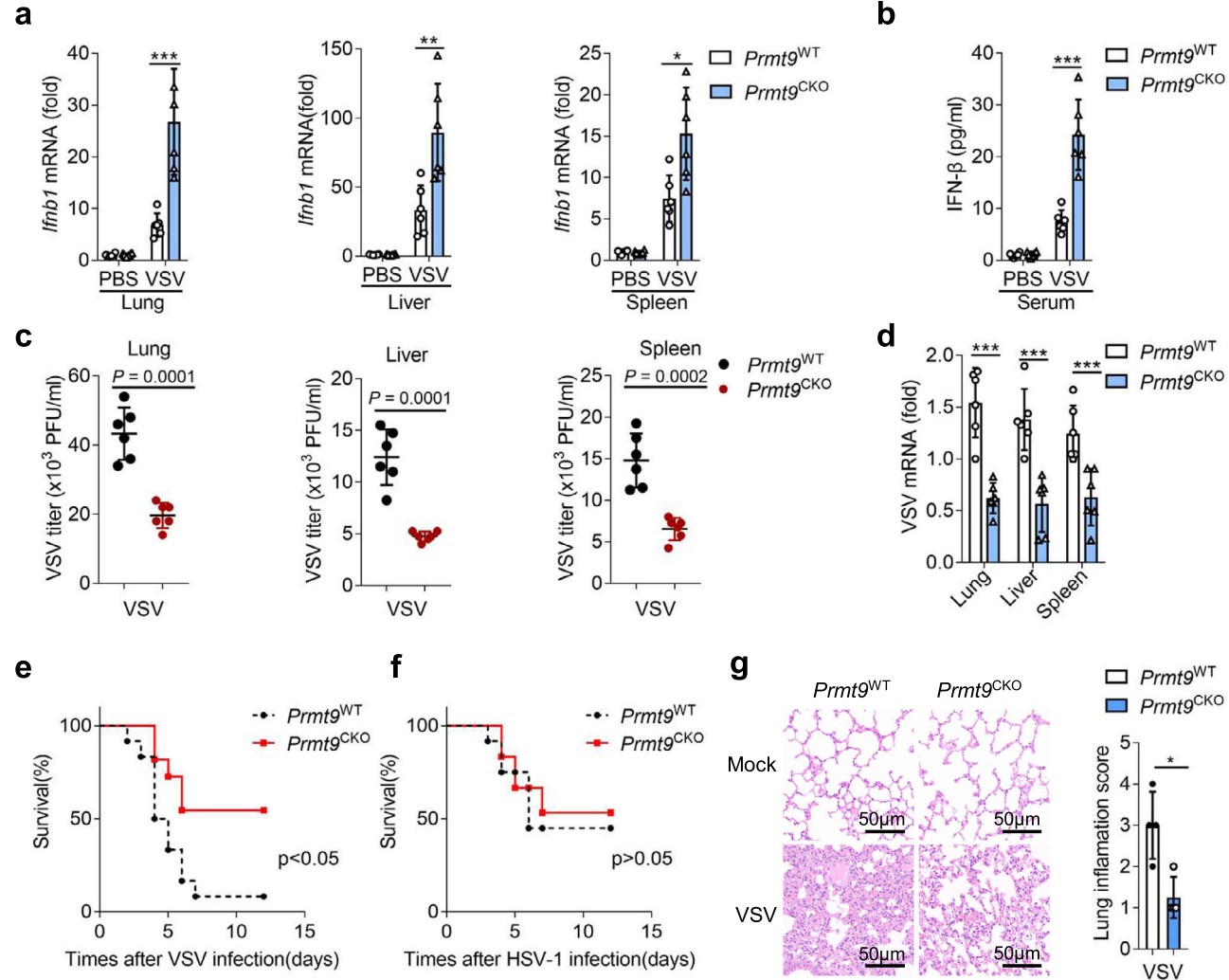

**Fig. 3 | PRMT9 inhibits innate antiviral response in vivo. a–d** Eight-week male *Prmt9*^CKO and *Prmt9*^WT mice were infected by tail vein injection with VSV (1.8 × 10^7 PFU per mouse) for 24 h (*n* = 6 mice per group). **a** qRT-PCR analysis of *Ifnb1* mRNA in the lung (left, ***p = 0.0009), liver (middle, **p = 0.0059), and spleen (middle, *p = 0.0119). **b** ELISA analysis of IFN-β protein in serum, ***p = 0.0002. Plaque assay of VSV titers (**c**) and qRT-PCR analysis of VSV mRNA (**d**) in lung (left, ***p = 0.0001), liver (middle, ***p = 0.0005) and spleen (right, ***p = 0.0028). **e, f** Survival of *Prmt9*^CKO and *Prmt9*^WT mice (*n* = 12 mice per group, 6–8 weeks old) after tail vein

injection with VSV (1 × 10^8 PFU per mouse) or HSV-1 (1.5 × 10^8 PFU per mouse). **e:** *p = 0.0133, **f:** ns = 0.7923. **g** Hematoxylin-eosin staining of lung sections were presented, treated as in **a** (*n* = 6 mice per group). Scale bar, 50 μm. Inflammation scores of lung tissue sections described in **g** (*p = 0.0106). The qRT-PCR and ELISA results are presented relative to those of untreated wild-type tissue cells (**a**, **b** and **d**). Data are shown as mean ± SD (**a–g**) and are representative of three independent experiments with similar results. *P < 0.05; **P < 0.01; ***P < 0.001 (two-tailed student's *t*-test in **a–d**, **g** or the log-rank Mantel-Cox test in **e**, **f**).

pulmonary fibrosis compared with the *Prmt9*^WT mice following VSV infection (Fig. 3g). Altogether, the results suggest that PRMT9 deficiency enhances the antiviral innate immune response in vivo against RNA virus.

To further investigate the in vivo function of PRMT9 antiviral immune responses, PRMT9 complete knockout mice were produced by crossing homozygous *Prmt9*^fl/fl mice with Cre-ERT2 mice. *Prmt9*^KO and *Prmt9*^WT mice were intravenously infected with VSV or HSV-1. Consistent with the function of PRMT9 in antiviral response in conditional PRMT9 knockout mice, the expression levels of *Ifnb1* in the lung, liver, spleen and the production of IFN-β in the serum of *Prmt9*^KO mice were significantly higher than in those from *Prmt9*^WT (Supplementary Fig. 8a). Moreover, the VSV mRNA (Supplementary Fig. 8b) and VSV titers (Supplementary Fig. 8c) in the lung, liver, spleen of *Prmt9*^KO mice were found to be significantly greater than in those from *Prmt9*^WT. Further, the results of hematoxylin-and-eosin staining showed that PRMT9 deficiency alleviated inflammatory cell infiltration, tissue edema and pulmonary fibrosis compared with the *Prmt9*^WT mice following VSV infection (Supplementary Fig. 8d). *Prmt9*^KO mice were also

found to be less susceptible than *Prmt9*^WT mice to infection with VSV (Supplementary Fig. 8g), while there were no differences in survival between *Prmt9*^KO mice and *Prmt9*^WT mice after infection with HSV-1 (Supplementary Fig. 8h). As shown in Supplementary Fig. 8e, f, the production of IFN-β protein and *Ifnb1* mRNA was barely affected by *Prmt9*-knockout. Consistent with the expression of *Ifnb1*, the results showed that the copy number of HSV-1 genomic DNA and the HSV-1 titers of HSV-1 were not affected in *Prmt9*^KO mice relative to that in *Prmt9*^WT mice. Overall, PRMT9 deficiency enhances the antiviral innate immune response in vivo against RNA virus, but not for HSV-1.

### Negative regulation of IRF3 and NF-κB-signaling pathway during viral infection

To investigate the action of PRMT9 in the regulation of MAVS-mediated signaling pathway, Flag-PRMT9 plasmids were transfected into HEK293T cells followed by infection with SeV. The results of immunoblot analysis showed that the levels of SeV-induced phosphorylation of TBK1, IRF3 and IκBα were decreased in HEK293T cells transfected with the PRMT9 expression plasmids compared with an empty vector

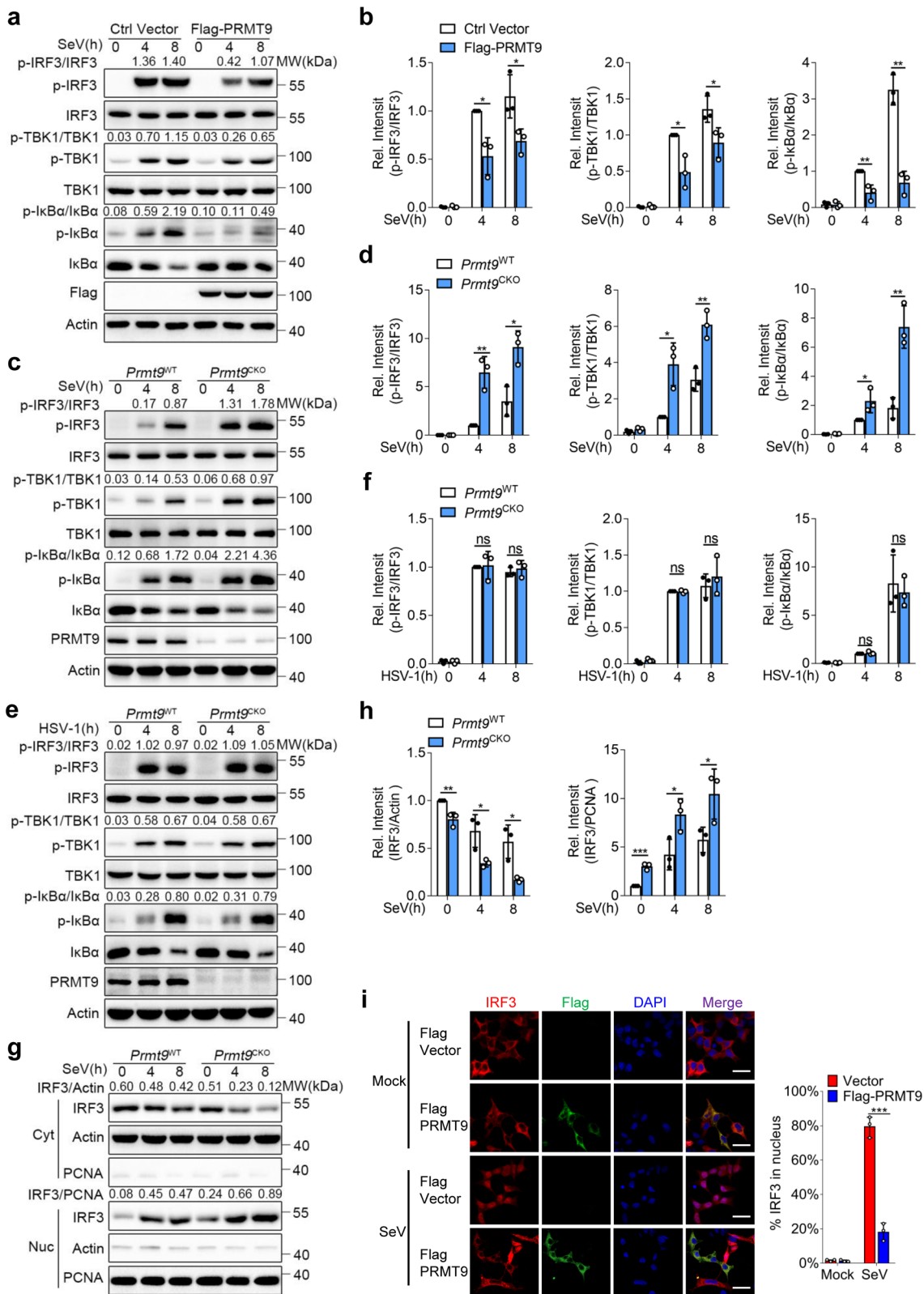

(Fig. 4a, b). We further prepared peritoneal macrophages from *Prmt9*[WT] and *Prmt9*[CKO] mice followed infection with SeV and HSV-1 or stimulation with 5'ppp RNA. We found that the levels of phosphorylation of TBK1, IRF3 and IκBα were higher in *Prmt9*[CKO] macrophages than those of *Prmt9*[WT] macrophages after infection with SeV or stimulation with 5'ppp-RNA (Fig. 4c, d and Supplementary Fig. 5i, j). The levels of phosphorylation of TBK1 and IRF3 were also significantly increased in

*Prmt9*-knockdown macrophages (Supplementary Fig. 5a–d) and *Prmt9*-knockout RAW cells after stimulation of SeV (Supplementary Fig. 5e, f). In contrast, the phosphorylation of TBK1, IRF3 and IκBα exhibited no differences in *Prmt9*[CKO] macrophages or *Prmt9*-knockout RAW cells after infection with HSV-1 (Fig. 4e, f and Supplementary Fig. 5g, h). As previously reported[9], after phosphorylation, IRF3 will dimerize and translocate into the nucleus. Therefore, the cytoplasmic and nucleic

**Fig. 4 | PRMT9 negatively regulates MAVS-mediated signaling. a** Immunoblot analysis of total and phosphorylated (p-) TBK1, total and phosphorylated (p-) IRF3, total and phosphorylated (p-) IκBα in lysates of HEK293T cells transfected with control vector or Flag-PRMT9 infected with SeV for 0–8 h. **c, e** Immunoblot analysis of total and p-TBK1, total and p-IRF3, total and p-IκBα in $Prmt9^{CKO}$ and $Prmt9^{WT}$ peritoneal macrophages cells, followed by infection with SeV or HSV-1 for 0-8 h. **b, d** and **f** Densitometric analysis of protein expression levels was quantitated by "ImageJ" software. Ratio: p-TBK1/TBK1, p-IRF3/IRF3 or p-IκBα/IκBα, bar graphs are presented relative to the second lane. Data are represented as mean ± SD (two-tailed student's $t$-test was performed, for **b**, left panel: $^*p = 0.0132, 0.0354$ in sequence, middle panel: $^*p = 0.0232, 0.0278$ in sequence, right panel: $^{**}p = 0.0082, 0.0010$ in sequence; for **d**, left panel: $^{**}p = 0.0044$, $^*p = 0.0126$ in sequence, middle panel: $^*p = 0.0129$, $^{**}p = 0.0061$ in sequence,

right panel: $^*p = 0.0486$, $^{**}p = 0.0041$ in sequence; $n = 3$ independent experiments). **g** Immunoblot analysis of IRF3 in cytoplasmic (Cyt) and nuclear (Nuc) in $Prmt9^{CKO}$ and $Prmt9^{WT}$ peritoneal macrophages cells infections with SeV for 0–8 h. **h** Densitometric analysis of protein expression levels, bands were normalized with individual actin or PCNA, bar graphs are presented relative to the lane 1. Data are represented as mean ± SD (two-tailed student's $t$-test $Prmt9^{WT}$ vs. $Prmt9^{CKO}$, left panel: $^{**}p = 0.0095$, $^*p = 0.0297, 0.0176$ in sequence, right panel: $^{***}p = 0.0005$, $^*p = 0.0341, 0.0476$ in sequence; $n = 3$ independent experiments). **i** Confocal microscopic images of IRF3 (Red) in HEK293T cells transfected with control vector or Flag-PRMT9 (Green) infected with SeV for 0–8 h, Scale bar, 10 μm, $^{***}p = 0.0008$. $^*P < 0.05$; $^{**}P < 0.01$; $^{***}P < 0.001$; ns, not significant (two-tailed student's $t$-test). Similar results were obtained from three independent experiments.

fractions were separated from SeV-infected macrophages, the IRF3 nuclear translocation was measured. The translocation of IRF3 to the nucleus was found to be greater in $Prmt9^{CKO}$ macrophages than that of $Prmt9^{WT}$ macrophages after infection with SeV (Fig. 4g, h). Furthermore, the results of confocal microscopy showed that overexpression of PRMT9 in HEK293T cells attenuated the translocation of IRF3 to the nucleus (Fig. 4i). Taken together, the data suggest that PRMT9 inhibits IFN-β production and innate antiviral immunity through inhibition of MAVS-mediated signaling.

## PRMT9 targets MAVS

To define the targets that are regulated by PRMT9, the overexpression plasmids of the main signaling proteins associated with the innate antiviral immunity were transfected into HEK293T cells together with the PRMT9-expressed plasmids and the IFN-β promoter luciferase reporter. We found that activation of the IFN-β promoter luciferase reporter mediated by RIG-I, MDA5 or MAVS were inhibited by PRMT9 overexpression (Fig. 5a). Meanwhile, the activation of the IFN-β promoter luciferase reporter mediated by TBK1 were not affected by PRMT9 overexpression (Fig. 5a). cGAS/STING-induced IFN-β promoter activation was not affected by PRMT9 overexpression (Fig. 5a), which is consistent with the data showing that PRMT9 could not regulate the antiviral immune response against DNA virus infection. Further findings were made that the levels of *IFNB1* expression mediated by RIG-I, MDA5 or MAVS in HEK293T cells were also inhibited by PRMT9 expression, while, the levels of *IFNB1* expression mediated by TBK1 or IRF3 were not impaired upon PRMT9 overexpression (Fig. 5b). Such data suggest that PRMT9 may target MAVS to regulate IFN-β signaling and antiviral response.

For further verification, HA-cGAS, HA-RIG-I, HA-MAVS, HA-TBKI, Myc-MDA5 and Myc-IRF3 were transfected together with GFP-PRMT9 into HEK293T cells to investigate the interactions thereof. The results of co-immunoprecipitation and western blotting analysis showed that PRMT9 interacted with MAVS but not with cGAS, RIG-I, MDA5, TBKI, and IRF3 (Fig. 5c). The results of in vitro pull-down with recombinant proteins showed the direct interaction between PRMT9 and MAVS (Fig. 5d). Immunofluorescence results also showed significant co-localization between PRMT9 and MAVS (Fig. 5e). Further, the interaction between endogenous PRMT9 and MAVS was found to be attenuated upon viral infection (Fig. 5f). Taken together, the data demonstrate that PRMT9 inhibited antiviral innate immunity by targeting MAVS on the mitochondria.

## PRMT9 promotes MAVS methylation

PRMT9 is a type II arginine methyltransferase, which can catalyze the monomethylation (MMA) or symmetrical dimethylation (SDMA) of arginine residues in target proteins. Protein arginine methylation is significant in a number of physiological functions[26]. As such, we investigated whether PRMT9 could catalyze MAVS arginine methylation. GFP-PRMT9 expression plasmids were first transfected into HEK293T cells together with HA-MAVS. The results of co-

immunoprecipitation (Co-IP) and western blotting assay showed that PRMT9 enhanced the SDMA level of MAVS (Fig. 6a, left) rather than PRMT5 and PRMT7 (Supplementary Fig. 4a, b). PRMT9 has been reported to catalyze SDMA of SAP145[27], and thus, HA-SAP145 plasmid was used to make for positive control (Fig. 6a, right). Since there are three types of arginine methylation are catalyzed by arginine methyltransferases[28], the type of methylation of MAVS was examined. The primary peritoneal macrophages were isolated from wide type mice and infected with SeV. The results of Co-IP and immunoblot analysis showed the formation of MMA and SDMA, but not asymmetrical dimethylation (ADMA) in MAVS without SeV infection (Fig. 6b). Notably, the formation of MMA and SDMA gradually decreased upon SeV infection (Fig. 6b). To further determine the methylated arginine function of PRMT9, the SDMA of endogenous MAVS were examined through Co-IP and western blotting. Consistent with the results of overexpression of PRMT9, knockout of PRMT9 in peritoneal macrophages from $Prmt9^{CKO}$ and $Prmt9^{WT}$ mice greatly decreased MAVS methylation (Fig. 6c). Similarly, siRNA knockdown of PRMT9 expression in macrophages also attenuated the formation of SDMA in macrophages (Supplementary Fig. 4c). Overall, the data demonstrate that PRMT9 regulated the methylation of MAVS in the RLRs pathway.

As reported in a previous study, mutation at Gly260 renders PRMT9 into a catalytically inactive form[29]. To verify whether PRMT9-mediated MAVS methylation depends on its methyltransferase activity, WT PRMT9 or PRMT9 mutant G260E were transfected together with MAVS into HEK293T cells. The results indicated that WT PRMT9 could increase the level of MAVS methylation, while the PRMT9 enzymatic mutant G260E lost the ability to induce MAVS methylation (Fig. 6d). SeV infection decreased PRMT9-mediated MAVS methylation (Fig. 6d). We also reintroduced WT PRMT9 and PRMT9 mutant G260E into $Prmt9^{CKO}$ macrophages and measured MAVS methylation. We found the expression of mPRMT9 in $Prmt9^{CKO}$ macrophages restored MAVS methylation, whereas the expression of mPRMT9 mutant G260E had no such effect (Fig. 6e). The data demonstrate that PRMT9 catalyzed MAVS methylation through the methyltransferase activity thereof.

To directly investigate PRMT9 catalyze arginine methylation on MAVS, recombinant proteins were prepared for His-MAVS and GST-PRMT9, and performed an in vitro protein arginine methylation assay. Findings were made that MAVS methylation was increased in the presence of PRMT9 in the in vitro methylation system (Fig. 6f). Taken together, these data demonstrate that PRMT9 promoted MAVS arginine methylation.

## PRMT9 catalyzes MAVS methylation at R41 and R43

To determine the potential arginine residues in MAVS that were modulated by PRMT9, we performed an in vitro methylation assay using recombinant MAVS and PRMT9 protein, then liquid chromatography–mass spectrometry (LC–MS) analysis was used to identify the arginine residues that were methylated. In the assay, R41 and R43 residues of MAVS were identified as potential methylation sites (Fig. 6g). Next, R41 and R43 arginine residues were mutated to

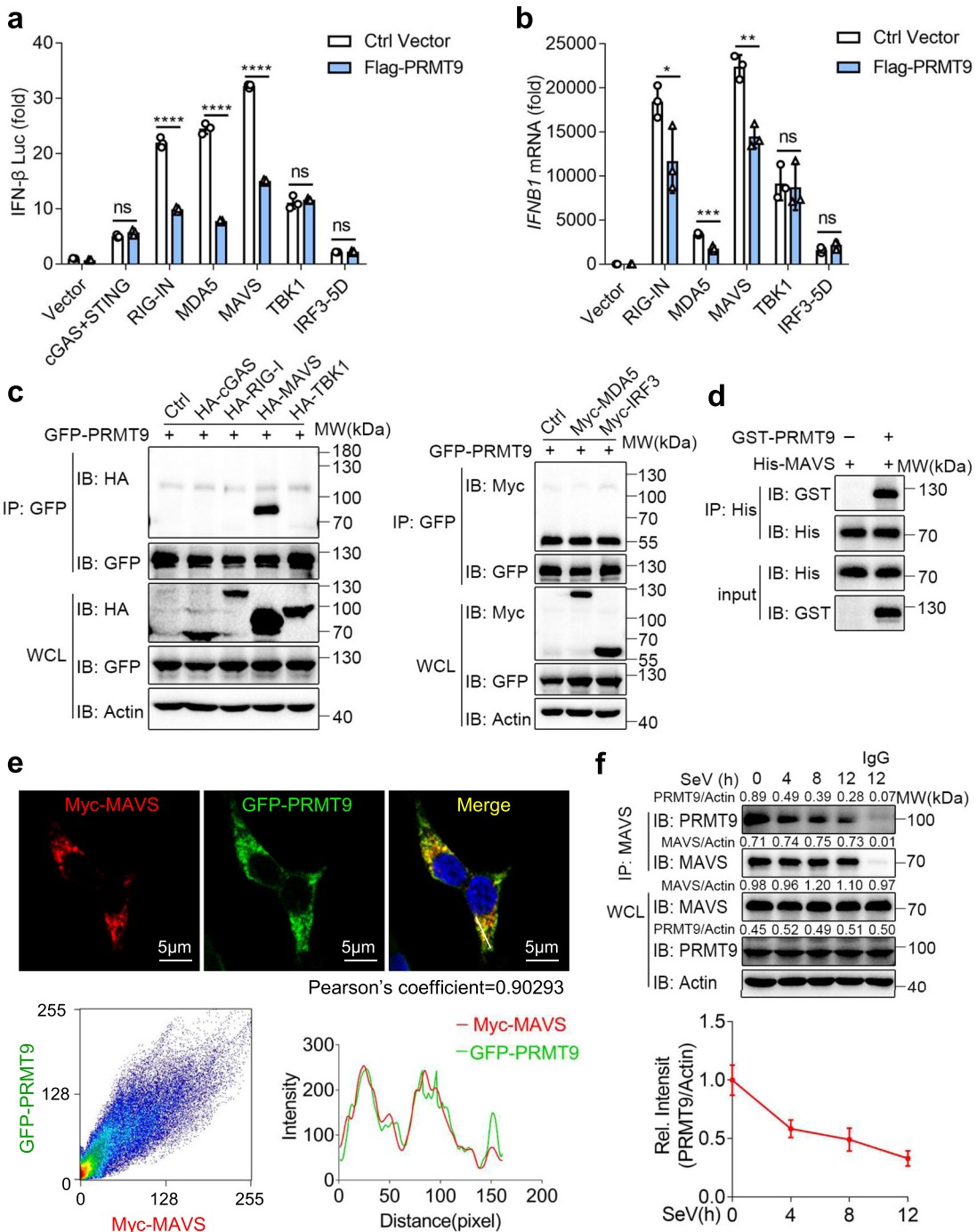

**Fig. 5 | PRMT9 targets MAVS. a** Luciferase (Lucif) activity assays of IFN-β reporter in HEK293T cells transfected with expression plasmids for cGAS-STING, RIG-IN, MDA5, MAVS, TBK1, or IRF3-5D along with Flag-PRMT9 or control vector (Ctrl) for 24 h (mean ± SD, two-tailed student's *t*-test Flag-PRMT9 vs. Ctrl Vector, ns = 0.0959, ****$p$ < 0.0001, ****$p$ < 0.0001, ****$p$ < 0.0001, ns = 0.5018, ns = 0.5552 in sequence; $n$ = 3 independent experiments). **b** qRT-PCR analysis of *IFNB1* mRNA in HEK293T cells transfected with the indicated adaptors with Flag-PRMT9 or control vector (mean ± SD, two-tailed student's *t*-test Flag-PRMT9 vs. Ctrl Vector, *$p$ = 0.0446, ***$p$ = 0.0008, **$p$ = 0.0018, ns = 0.8231, ns = 0.1474 in sequence; $n$ = 3 independent experiments). **a, b** Results are presented relative to those of untreated cells transfected with a Vector plasmid. **c** Co-IP analysis of the interaction between PRMT9 and adaptors in HEK293T cells cotransfected with GFP-PRMT9 and HA-cGAS, HA-RIG-I, HA-MAVS, HA-TBK1, Myc-MDA5 or Myc-IRF3 in HEK293T cells plasimds. **d** In vitro analysis of the interaction between PRMT9 and MAVS, using recombinant protein GST-PRMT9 and His-MAVS incubated in vitro. **e** Confocal analysis of the colocalization of Myc-MAVS (Red) and GFP-PRMT9 (Green) in HEK293T cells cotransfected for 24 h. Scale bars: 5 µm. Intensity profiles of each line was quantified by ImageJ software and drawn by GraphPad Prism 7.0. Myc-MAVS - GFP-PRMT9 colocalization was also quantified using Pearson's correlation coefficient method and scatter map. **f** Co-IP analysis of the interaction between PRMT9 and MAVS in mouse peritoneal macrophages infected with SeV for 0–12 h. Densitometric analysis of protein expression levels, bands were normalized with individual actin, line graphs are presented relative to the first lane. *$P$ < 0.05; **$P$ < 0.01; ***$P$ < 0.001; ns, not significant (Two-tailed Student's *t*-test). Similar results were obtained from three independent experiments.

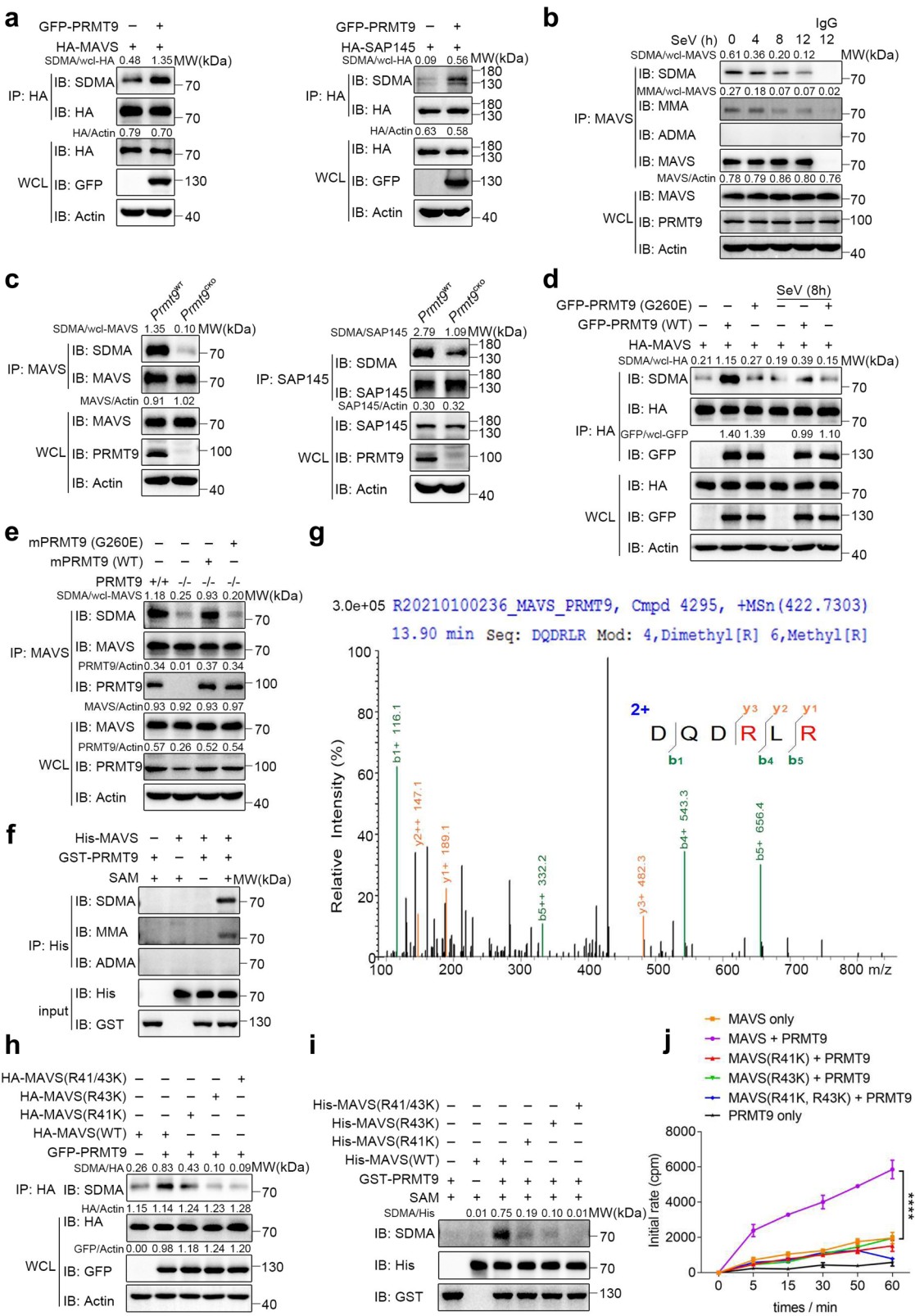

lysine, and the methylation status of MAVS in MAVS KO HEK293T cells (*MAVS*$^{-/-}$) was examined. We found PRMT9 could promote the methylation of MAVS (WT), while, MAVS (R41K), MAVS (R43K) and MAVS (R41K, R43K) mutation led to significant reductions in MAVS methylation in the presence of PRMT9 (Fig. 6h). Recombinant MAVS proteins were prepared for MAVS (R41K), MAVS (R43K) and MAVS (R41K, R43K), and an in vitro methylation assay was performed. The

results of western blotting showed that PRMT9 could promote MAVS (WT) methylation, while PRMT9-mediated MAVS methylation was greatly decreased in MAVS (R41K), MAVS (R43K) and MAVS (R41K, R43K) (Fig. 6i). Further, an in vitro methylation assay was performed with *S*-[$^3$H-Met] adenosylmethionine as the methyl donor. The results showed that the radioactivity of His-MAVS was greatly increased in the presence of recombinant PRMT9 protein (Fig. 6j), while radioactivity

**Fig. 6 | PRMT9 catalyzed the arginine methylation of MAVS on R41 and R43.** **a**–**f** Co-IP analysis of the methylation of MAVS. **a** In HEK293T cells cotransfected with GFP-PRMT9 and HA-MAVS or SAP145 (as positive control). **b** In peritoneal macrophages infected with SeV for 0–12 h, and **c** in *Prmt9*CKO and *Prmt9*WT peritoneal macrophages cells. **d** Co-IP analysis of the SDMA of MAVS in *MAVS*−/− HEK293T cells cotransfected with HA-MAVS and GFP-PRMT9 (WT) or PRMT9 (G260E) for 24 h, followed by infection with SeV for 8 h. **e** Co-IP analysis of the methylation of MAVS in *Prmt9*CKO and *Prmt9*WT peritoneal macrophages cells reconstituted with empty vector (− or Vec) or plasmids expressing GFP-tagged mouse PRMT9 (WT) or mouse PRMT9 (G260E). **f** PRMT9 methylates MAVS directly. The in vitro MAVS methylation assay was performed by using recombinant GST-PRMT9 and His-MAVS proteins in the presence of SAM or not. **g** Mass spectrometry profile of methylated MAVS-R41, R43 sites. **h** Co-IP analysis of of SDMA of MAVS in *MAVS*−/− HEK293T cells cotransfected with HA-MAVS (WT) or HA-MAVS (R41K), HA-MAVS (R43K), HA-MAVS (R41, 43K) in the presence of GFP-PRMT9 expression plasmids. **i** In vitro MAVS methylation assay using recombinant GST-PRMT9, His-MAVS and His-MAVS muntans proteins the presence of PRMT9 or not. **j** PRMT9 methylates MAVS at R41/43 in vitro. The scintillation counting assay was performed by using purifed GST-PRMT9, His-MAVS (WT), His-MAVS (R41K), His-MAVS (R43K), His-MAVS (R41K, R43K). Methyltransferase activity was monitored by the transfer of $^3$H-methyl (3H-Me) from S-[3H-Me] adenosylmethionine to recombinant protein substrates. After exposure at 30 °C for the indicated time, Reaction mixtures were spotted onto Whatman 3 mm cellulose filter paper discs for analyzing. The data are representative of three independent experiments with similar results (**a**–**f**, **h**, **i**) or are from three independent experiments (mean of triplicate assays **j**, ****$P < 0.0001$, two-way ANOVA).

levels of MAVS (R41K), MAVS (R43K) and MAVS (R41K, R43K) were significantly decreased compared with that of MAVS (WT) (Fig. 6j). Taken together, these data suggest that PRMT9 catalyzed the arginine methylation of MAVS on R41 and R43 residues.

## PRMT9-mediated MAVS methylation inhibits MAVS aggregation and activation

PRMT9 catalyzed the arginine methylation of MAVS, and PRMT9 negatively regulated RLRs-mediated innate antiviral signaling. Therefore, we hypothesized that MAVS arginine methylation mediated by PRMT9 may inhibit MAVS activation and the RLRs-induced signaling. Consistent with the hypothesis, PRMT9 enzymatic mutant G260E (PRMT9 (G260E)) was found to have lost the ability to decrease SeV-induced *IFNB1* expression in HEK293T cells compared with PRMT9 (Supplementary Fig. 6a). Similarly, VSV infection-induced expression of *IFNB1* mRNA was not attenuated by PRMT9 (G260E) compared with PRMT9 in HEK293T cells (Supplementary Fig. 6b). Consistently, VSV replications as measured by VSV titers and VSV mRNA were not significantly potentiated by PRMT9 (G260E) compared with cells that expressed the Vector (Supplementary Fig. 6b). We also reintroduced PRMT9 (WT) and PRMT9 (G260E) into *Prmt9*CKO macrophages, followed by infection with SeV and VSV. The data showed that PRMT9 (WT), but not PRMT9 (G260E), restored the suppression of SeV- and VSV-induced phosphorylation of TBK1 and IRF3 (Supplementary Fig. 6c), which indicated that the negative regulation of RLRs signal by PRMT9 was dependent on the methyltransferase activity thereof.

RNA virus infection induces MAVS to rapidly form prion-like aggregates which lead to the robust induction of type I IFNs and other inflammatory cytokines[7,30]. In the present study, an investigation was conducted into whether PRMT9-mediated MAVS methylation inhibits MAVS aggregation. GFP-PRMT9 expression plasmid or GFP-PRMT9 (G260E) and HA-MAVS was first transfected into HEK293T cells, and MAVS aggregation was measured through Semi-denaturing detergent agarose-gel electrophoresis (SDD-AGE). Overexpression of MAVS could promote the formation of MAVS aggregates as previously reported (Fig. 7a)[15]. Consistent with the inhibitory effect of PRMT9, the aggregation of MAVS was found to be decreased in PRMT9 overexpressed HEK293T cells than that of cells transfected with the control vector (Fig. 7a). Notably, overexpression of PRMT9 (G260E) could not decrease MAVS aggregation (Fig. 7a). We further measured MAVS aggregation induced by SeV infection in peritoneal macrophages prepared from *Prmt9*CKO and *Prmt9*WT mice. We found that SeV infection could induce MAVS aggregation in macrophages (Fig. 7b), while PRMT9 deficiency greatly increased the formation of MAVS aggregates in *Prmt9*CKO macrophages after infection (Fig. 7b). Notably, a low level of MAVS aggregation could be detected in *Prmt9*CKO macrophages even without SeV infection (Fig. 7b), indicating PRMT9-mediated MAVS methylation could prevent MAVS from autonomous activation under normal conditions.

To further verify whether the formation of MAVS aggregation was regulated by PRMT9 and dependent on the enzymatic activity thereof,

we reintroduced PRMT9 (WT) and PRMT9 (G260E) into *Prmt9*CKO macrophages, the data showed that the reintroduction of PRMT9 (WT), but not PRMT9 (G260E) attenuated the SeV-induced formation of MAVS aggregates in *Prmt9*CKO macrophages (Fig. 7c, lane3). Consistent with such data, PRMT9 (WT) inhibited the expression of *Ifnb1* mRNA under stimulation of SeV or VSV, but not PRMT9 (G260E) in *Prmt9*CKO macrophages (Fig. 7d).

To investigate the function of PRMT9-induced methylation in the activation and aggregation of MAVS, wild-type MAVS (MAVS(WT)) or MAVS arginine methylation site mutant MAVS (R41K, R43K) plasmids were transfected into *MAVS*−/− HEK239T cells together with PRMT9 expression plasmids. The data showed that PRMT9 could induce the methylation of MAVS (WT), whereas MAVS (R41K, R43K) led to a significant reduction in methylation level mediated by PRMT9 (Fig. 7e). Notably, we found that expression of PRMT9 inhibited MAVS aggregation in *MAVS*−/− HEK239T cells transfected with the MAVS (WT) expressing plasmid, but not in MAVS (R41K, R43K) transfected cells (Fig. 7f). Consistent with the MAVS aggregation data, SeV-induced activation of *IFNB1*, *IFNA4*, and *CXCL10* was inhibited only in *MAVS*−/− HEK239T cells transfected with the MAVS (WT) expressing plasmid, but not MAVS (R41K, R43K) in the presence of PRMT9 (Fig. 7g). The results suggest that PRMT9-induced methylation inhibits MAVS aggregation and downstream signaling.

To confirm that PRMT9-induced arginine methylation inhibits MAVS aggregation directly, we isolated crude mitochondria (P5) from *MAVS*−/− HEK293T cells transfected with MAVS (WT) or MAVS (R41K, R43K), then performed in vitro MAVS aggregation assay. The mitochondria fraction was incubated with full-length RIG-I protein, 5′-pppRNA and unanchored K63-linked ubiquitin chains in a cell-free system. As reported[7], the addition of RIG-I, 5′-pppRNA and unanchored K63-linked ubiquitin chains could induce MAVS aggregation as measured by SDD-AGE (Fig. 7h). However, the formation of MAVS aggregation was greatly decreased in the presence of PRMT9 (Fig. 7h). Consistent with the data that PRMT9-mediated MAVS methylation inhibits MAVS aggregation, PRMT9 could not prevent MAVS (R41K, R43K) from forming MAVS aggregates (Fig. 7h). Overall, these data indicate that PRMT9 promoted MAVS methylation to inhibit MAVS aggregation and activation.

## Discussion

MAVS is the key adapter protein of the RLRs signaling pathway involved in the host defense against various RNA viruses[31,32]. As is well known, MAVS activation is tightly coupled to the aggregation thereof, which can only be formed following RNA virus infection[33]. To maintain immune homeostasis, the formation of spontaneous aggregation and activation of MAVS should be finely tuned to protect the host from an overactive immune response, especially in the resting state. In the present study, PRMT9 promoted the modification of SDMA on MAVS and inhibited the aggregation thereof in the resting state. Therefore, the present research reports a PTM of MAVS, which has an essential function in keeping MAVS inactive under physiological conditions.

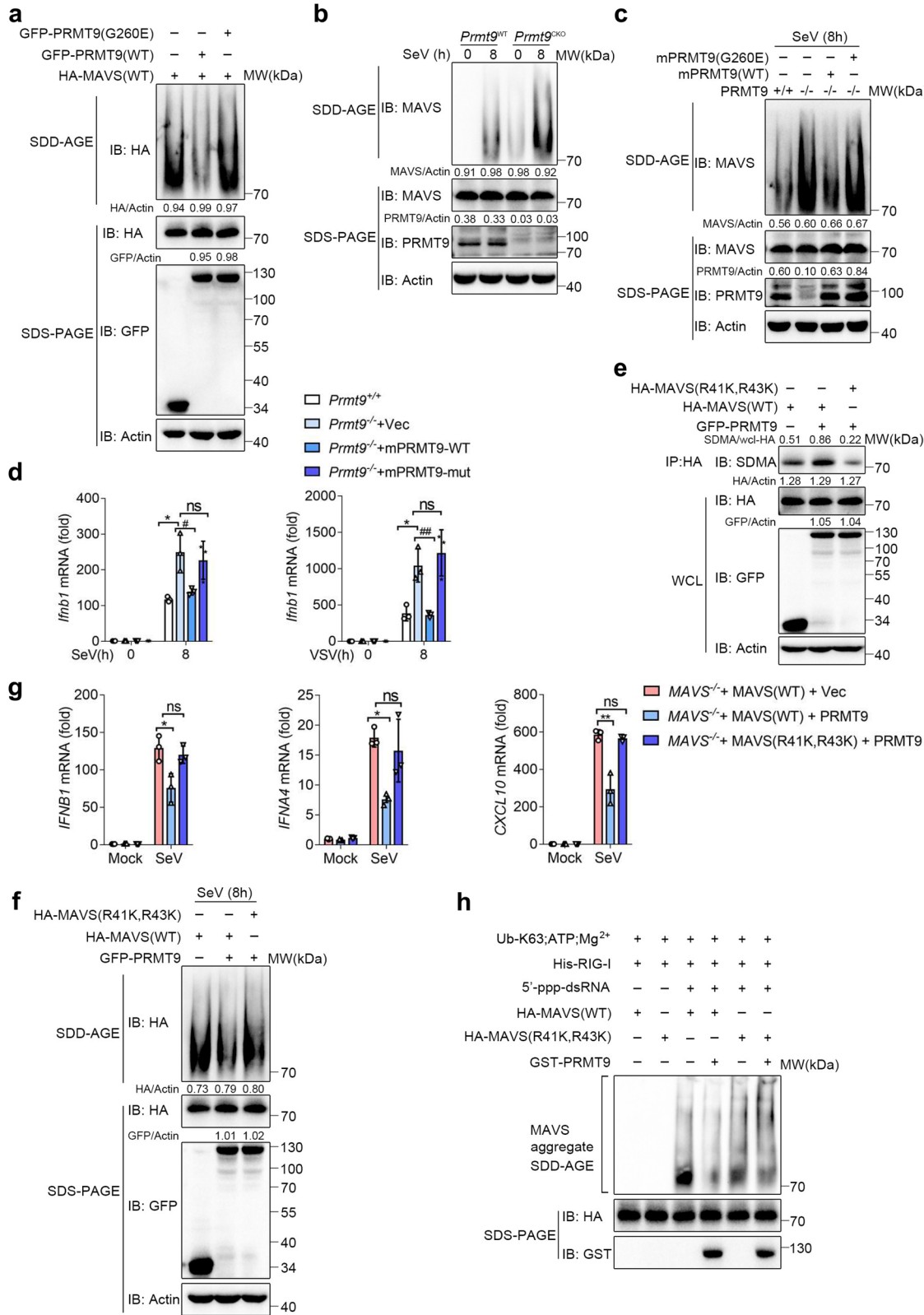

PRMT9 presents an effective regulator that prevents endogenous MAVS from spontaneous aggregation.

PRMT9 is a type II Protein arginine methyltransferase, which catalyzes the symmetric arginine dimethylation of the target protein[29,34,35]. The catalytic activity of PRMT9 was unknown and named FBXO11 until Yang et al. identified it as the second SDMA-forming enzyme in mammals[27,29]. Several studies have shown that FBXO11 was previously

erroneously named PRMT9[27,29,36]. A recent study demonstrates that PRMT9 has two distinguishing features: AdoMet-binding motif, and three N-terminal tetratricopeptide repeats (TPR), a hallmark amino acid sequence motif belonging to the PRMT family. Studies have proved that FBXO11 does not contain the PRMT family distinguishing features[37]. In addition, the nematode homolog of FBXO11 has been observed no methyltransferase activity[37]. Therefore, researchers

**Fig. 7 | PRMT9-mediated MAVS methylation inhibits MAVS aggregation and activation. a** SDD-AGE analysis (top) and SDS-PAGE (below) of the aggregation of MAVS in HEK293T cells cotransfected with HA-MAVS, GFP-PRMT9 and PRMT9 (G260E). **b** SDD-AGE and SDS-PAGE analysis of the aggregation of MAVS in *Prmt9*[CKO] and *Prmt9*[WT] peritoneal macrophages cells infected with SeV for 0–8 h. **c** SDD-AGE and SDS-PAGE analysis of the aggregation of MAVS in *Prmt9*[CKO] and *Prmt9*[WT] cells reconstituted with empty vector (− or Vec) or mPRMT9 (WT) or mPRMT9 (G260E) infected with SeV for 8 h. **d** qRT-PCR analysis of *Ifnb1* mRNA in *Prmt9*[CKO] and *Prmt9*[WT] reconstituted with empty vector or mPRMT9 (WT) or mPRMT9 (G260E) infected with SeV or VSV for 8 h, mRNA results are presented relative to those of untreated cells transfected with a Vector plasmid (Average of three replicates), left panel: *$P$ = 0.0130, #$P$ = 0.0241, ns = 0.6343 in sequence; right panel: *$P$ = 0.0105, ##$P$ = 0.0070, ns = 0.4865 in sequence. **e** Co-IP analysis of SDMA of MAVS in *MAVS*[−/−] HEK293T cells cotransfected with HA-MAVS (WT) or HA-MAVS (R41K, R43K) in the presence of GFP-PRMT9 expression plasmid. **f** SDD-AGE and SDS-PAGE analysis of the aggregation of MAVS in *MAVS*[−/−] HEK293T cells transfected with HA-MAVS (WT) or HA-MAVS (R41K, R43K) in the presence of GFP-PRMT9 expression plasmid infected with SeV for 8 h. **g** qRT-PCR analysis of *IFNB* (*$P$ = 0.0224, ns = 0.4906 in sequence), *IFNA4* (***$P$ = 0.0006, ns = 0.5296 in sequence) and *CXCL10* (**$P$ = 0.0043, ns = 0.3199 in sequence) mRNA treated as in **f**, mRNA results are presented as in **d**. **h** SDD-AGE and SDS-PAGE analysis of the aggregation of MAVS with the crude mitochondrial extracts prepared from *MAVS*[−/−] HEK293T cell after HA-MAVS and HA-MAVS (R41K, R43K) transfection, followed by incubation in the in vitro aggregation buffer with E1, 5′ppp RNA, GST-PRMT9 and ubiquitin. qRT-PCR data in **d**, **g** represent the mean ± SD ($n$ = 3 independent experiments). The data shown in **a**–**c** and **e**–**h** are from one representative experiment of at least 3 biological independent experiments. *$P$ < 0.05, **$P$ < 0.01, #$P$ < 0.05, ##$P$ < 0.01, ns, not significant (Two-tailed Student's $t$-test).

believe that the arginine methyltransferase activity of FBXO11 was most likely the result of contamination of the FLAG-tagged protein with PRMT5[38,39]. Several studies have shown that PRMT9 is highly expressed in several types of cancer (melanoma, and testicular, pancreatic and lymphoma cancer)[25,35]. Although PRMT9 has been linked to diverse cancer diseases, the function of PRMT9 in antiviral innate immunity remains unknown. The results of the present study indicate that PRMT9 negatively regulated RLRs-induced IFN-β production, both in vitro and in vivo. Notably, PRMT9 was shown to be an arginine methyltransferase that could directly bind to MAVS and inhibit the function thereof to limit the activation of RLRs signaling. Overexpression of PRMT9 significantly decreased IFN-β expression and facilitated viral replication, whereas knockdown or knockout of PRMT9 had the opposite effects upon viral infection. Compared with wide-type mice, PRMT9 deficient mice were less susceptible to infection with an RNA virus. Notably, PRMT9 (G260E) could not regulate the production of RLRs-mediated IFN-β. Therefore, we discovered a significant mechanism of PRMT9 in RLRs-mediated innate immunity.

MAVS can be regulated at post-translational level during viral infection. Evidently, the PTMs including ubiquitination, phosphorylation, O-GlcNAcylation and de ubiquitination have important functions in tightly regulating the expression and function of MAVS[40]. In the present study, a PTM of MAVS mediated by PRMT9 was identified. PRMT9 directly bound with MAVS and promoted MAVS methylation on Arg41 and Arg43 residues, which efficiently attenuated the MAVS-mediated antiviral immune response. Overexpression of PRMT9 was found to significantly induce MAVS SDMA modification, and the recombinant PRMT9 proteins could also promote MAVS methylation using the in vitro methylation assay. However, MAVS (R41K, R43K) mutant could not be methylated by PRMT9. Thus, these data showed that MAVS was methylated by PRMT9 and the Arg41 and Arg43 residue were the direct catalyzing sites of PRMT9.

As previously reported, MAVS has three domains: a C-terminal transmembrane (TM) domain, a middle proline-rich region (PRR) and an N-terminal caspase recruitment domain (CARD). During viral infection, N-terminal CARDs of MAVS promptly convert other MAVS on the mitochondrial outer membrane into prion-like aggregates[40]. In the present study, PRMT9-mediated methylation catalyzing site (Arg41 and Arg43) was found to be located on the CARD domain. PRMT9 transferred the methyl group (-CH$_3$) of S-adenosylmethionine (SAM) to MAVS, and thus, -CH$_3$ replaced a hydrogen atom on ω-NG of arginine[41]. The SDMA on the substrates might affect the hydrogen bond structure to disturb protein interactions[42]. Therefore, we speculated that the methylation of MAVS inhibited its aggregation through impairing heterotypic CARD−CARD interactions.

MAVS activation is tightly coupled to the aggregation thereof, which can only form aggregates following stimulation in vivo[33]. However, the mechanisms preventing MAVS from spontaneous aggregation or activation in the absence of stimulus in cells are not clear. Qi et al. reported that the truncated isoforms of MAVS can autoinhibit to limit the spontaneous aggregation and activation through homotypic interaction thereof[33]. In the present study, we found that PRMT9 catalyzes the arginine methylation of MAVS on Arg41 and Arg43 and this modification inhibited MAVS aggregation and autoactivation. Hence, we rationalized that PRMT9 could inhibit the autoactivation of MAVS in unstimulated cells, which may be a potential mechanism for a host to avoid autoimmune diseases. It has been reported that MAVS can be regulated by PRMT7, which promotes MAVS monomethylation and thus negatively regulates antiviral innate immunity[43]. Consistent with PRMT9, MAVS monomethylation catalyzed by PRMT7 can also inhibit MAVS aggregation and activation. However, the regulatory mechanism of MAVS aggregation mediated by PRMT7 and PRMT9 is different. The mechanism of PRMT7 regulated MAVS aggregation depends on the inhibition of TRIM31-triggered K63-linked polyubiquitination of MAVS, as well as suppression of MAVS/RIG-I interaction. In the present study, we demonstrated that PRMT9 directly inhibited MAVS aggregation and subsequent activation by catalyzing SDMA of MAVS, using the in vitro methylation assays and MAVS aggregation assays (Figs. 6f, and 7h). Notably, PRMT9-mediated suppression of MAVS autoactivation was broken during viral infection. The data showed that PRMT9 located on mitochondria in unstimulated cells, and dissociated from the mitochondria upon SeV infection (Supplementary Figs. 1 and 2). The decrease in the methylation level on baseline MAVS with the RNA virus stimulation, which led to the activation and aggregation of MAVS, might be caused by the PRMT9 cellular trans-localization. Same phenomenon that MAVS MMA was decreased in response to virus infection is reported by Zhu J. et al.[43]. However, the article offers different explanations for this phenomenon. Zhu J. et al. indicates that MAVS recruited SMURF1 to PRMT7 to catalyze the K48-linked polyubiquitination of PRMT7 upon RNA virus infection, which leads to the proteasomal degradation of PRMT7 and subsequent MMA attenuation of MAVS. Thereby, the decrease in MAVS methylation facilitated the formation of prion-like MAVS aggregates and the innate antiviral immune response activation.

In summary, the results of the present study demonstrate that PRMT9 targeted MAVS directly and catalyzed the arginine methylation of MAVS at the Arg41 and Arg43 residues and further attenuated MAVS-mediated antiviral innate immune response. We demonstrate that methylation of MAVS by PRMT9 can inhibit MAVS autoactivation to keep MAVS inactive under physiological conditions to avoid excessive harmful immunity (Supplementary Fig. 9).

## Methods
### Reagents
Antibodies were obtained as follows: anti-SDMA-Arginine (13222 S, 1:500); anti-ADMA -Arginine (13522 S, 1:500); anti-MMA-anti- Arginine (8015 S, 1:500); anti-IRF3 (4302 S, 1:1000); anti-p-IRF3 (4947 s, 1:1000); anti-TBK1(3013 S, 1:1000); anti-p-TBK1 (5483 S, 1:1000); and anti-PCNA (#13110, 1:1000) were purchased from Cell Singling Technologies; anti-PRMT9 (Abclonal, WG-03137D, 1:1000); anti-TOMM20 (Abcam,

ab186734, 1:1000); anti-VSV G (Sigma Aldrich, 1:500); anti-Myc (Origene, 9E10, 1:1000); anti-HA (Origene, TA180128, 1:1000); anti-Flag (Sigma Aldrich, F1804, 1:1000); anti-MAVS (Santa Cruz Biotechnology, sc-365333, 1:500 for WB, 1:100 for Co-IP); anti-GFP (Santa Cruz Biotechnology, sc-9996, 1:1000); anti-β-actin (ZSGB-BIO, TA-09 Mouse, 1:1000), Goat-anti-mouse AlexaFluor-488(Cat.#:A-11001; AB_2534069, 1:200), Goat-anti-rabbit AlexaFluor-488 (Cat.#:A-11034; AB_2576217, 1:200), Goat-anti-mouse AlexaFluor-568 (Cat.#:A-11004; AB_2534072, 1:200), Goat-anti-rabbit AlexaFluor-568 (Cat.#:A-11011; AB_143157, 1:200), and 5'-pppRNA were purchased from Invivogen, with 0.5 μg/ml being used as a final concentration.

## Mice

The *Prmt9*<sup>fl/fl</sup> mice were constructed by Cyagen using the CRISPR–Cas9 gene-editing system. Lyz2-Cre mice were obtained from Jackson Laboratory. *Prmt9*<sup>fl/fl</sup> mice were crossed with Lyz2-Cre mice to generate *Prmt9*<sup>fl/fl</sup> Lyz2-Cre mice. The Genotyping of *Prmt9*<sup>fl/fl</sup> was confirmed by means of PCR using the following primers, forward primer 5'-GGTTCTCTTCTTCCATCAAGTAG-3', reverse primer 5'-TCTTCCTTAAA-TACTTCCTCCGTG-3. The primers for Lyz2-Cre transgenic mice genotyping were oIMR3066 mutant 5'-CCCAGAAATGCCAGATTACG-3', oIMR3067 common 5'-CTTGGGCTGCCAGAATTTCTC-3' and oIMR3068 WT 5'-TTACAGTCGGCCAGGCTGAC-3'. All the mice used in the present study were on the C57BL/6 background. All animal experiments were based on the general guideline principles provided by the Association for Assessment and Accreditation of Laboratory Animal Care, and performed with approval from Scientific Investigation Board of the Medical School of Shandong University.

Homozygous tamoxifen-inducible Cre recombinase-estrogen receptor T2 mice (Cre-ERT2) were obtained from Dr. Hui Xiao (Institute Pasteur of Shanghai, CAS, Shanghai, China). The homozygous *Prmt9*<sup>fl/fl</sup> were crossed with Cre-ERT2 mice to generate CreERT *Prmt9*<sup>fl/fl</sup> mice. CreERT *Prmt9*<sup>fl/fl</sup> mice were injected with tamoxifen (0.1 ml, 10 mg/ml in corn oil) by means of intraperitoneal injection once every 24 h for a total of 5 consecutive days to induce PRMT9 complete knockout[44]. Following the final injection, another 7 days must be wait before histological analysis to ensure Cre characterization work.

## Cells and viruses

Peritoneal macrophages were prepared as described previously[15]. THP-1 was cultured in 1640 basic medium supplemented with 15% FBS, and 1% penicillin-streptomycin, whereas HEK293T, HEK293 and Raw264.7 cells were cultured in high glucose DMEM supplemented with 10% FBS, 1% penicillin-streptomycin. All cells were cultured at 37 °C under 5% CO₂.

SeV was purchased from the China Center for Type Culture Collection (Wuhan University, China). Vesicular stomatitis virus (VSV), VSV-GFP and HSV-1 were offered by H. Meng (Institute of Basic Medicine, Shandong Academy of Medical Sciences, China).

## Generation of PRMT9 or MAVS KO cells

CRISPR-Cas9 gene-editing system was used to generate *Prmt9*-knockout Raw264.7 cell lines and *MAVS*- knockout HEK293T cells[45,46]. Firstly, a guide RNA (sgRNA) we designed based on the 'http://www.genome-engineering.org'. The sgRNA target sequences of PRMT9 (Mus, NM_001081240.3) and MAVS (Human, Gene ID 57506) were as follows:

Oligo F 5'- CACCGTGGAACTTCCGGTCGGCTTG-3',
Oligo R 5'- AAACCAAGCCGACCGGAAGTTCCAC-3' for MAVS.
Oligo F1 5'- CACCGACTTTTACCGCGTAGCAAAC-3',
Oligo R1 5'- AAACGTTTGCTACGCGGTAAAAGTC-3' for PRMT9 #1
Oligo F2 5'- CACCGCCGTTCTGGATATCGGCACG-3'
Oligo R2 5'- AAACCGTGCCGATATCCAGAACGGC-3' for PRMT9 #2

Secondly, the sgRNA was cloned into a lenti-CRISPRV2 vector, and then the cloned product was transformed into *E. coli* DH5α competent cells.

Thirdly, the successful plasmids were transiently transfected into HEK293T cells. The cells were cultured in DMEM supplemented with 10% FBS, and 1% penicillin-streptomycin, and all cells were cultured at 37 °C under 5% CO₂. For PRMT9, the virus supernatant was harvested after 48 h transfection, and stored at −80 °C. However, for MAVS, 48 h after transfection the cells were selected by medium supplemented with Puromycin (Sigma) 3ug/ml. Fourth, Raw264.7 cells were cultured as described above. Cells were transduced with lentivirus at 50–60% confluency. 48 h later, the cell cells were selected by media supplemented with Puromycin (Sigma) 3 ug/ml day by day. Finally, monoclonal cells were selected and identified.

## Plasmids, siRNA and transfection

Human PRMT9 cDNA (NM_138364.3), Human PRMT5 cDNA (NM_006109.4), and Human PRMT7 cDNA (NM_019023.3) were subcloned into a pCMV2-Flag vector. Wild-type GFP-PRMT9 were cloned into pBacPAK vector (Clontech) for insect cell expression. Human GFP-PRMT (1–9) were kindly provided by Yiping Wang (Fudan University, China), and pDsRED2-Mito was kindly provided by Dr. Jian Li (Shandong University). Flag-MAVS or HA-MAVS mutants (R41K, R43K, and R41/43K) and PRMT9 mutants (G260E) were generated by the QuikChange site-directed mutagenesis kits. cDNA of Human MAVS or MAVS mutants (Gene ID: 57506) were cloned into the vector pET30 (a). All plasmids were identified by means of DNA sequencing. Other plasmids used in this study came from our laboratory. Small interfering RNAs (siRNAs) against mPRMT9, mPRMT7, mPRMT5 or human PRMT9 were designed by GenePharma. The transient silencing target sequences are listed in Supplementary Table 1. Plasmids were transfected into HEK293T cells or HELA cells by Lipofectamine 3000 reagents (Thermo). siRNA duplexes were transiently transfected into macrophages or THP-1 cells using Lipofectamine RNA iMAX (Thermo).

## RNA quantification and ELISA

In brief, total RNA was extracted with the RNA fast200 kit (Fastagen) from cells, and reverse transcription of RNA was carried out using Reverse Transcription kit (Takara). Quantitative real-time RT-PCR was performed to analyze gene expression by using SYBR RT-PCR kit (Roche). The data were collected by qPCRsoft 4.0 (Bio-RAD). Data obtained for each gene was normalized to the expression of *β-Actin*. Gene-specific primers used for real-time RT-PCR are listed in Supplementary Table 2. ELISA kits for IFN-β were used to detect the concentrations of IFN-β culture supernatants and sera (R&D Systems). Elisa data were collected by Infinite M200 (Pro, Tecan, Switzerland).

## Luciferase reporter assay

The IFN-β promoter firefly luciferase reporter plasmid, a TK-Renilla luciferase reporter, together with Vector/PRMT9 alone or MAVS mutants, or other Plasmids were co-transfected into HEK293 cells. The cells were harvested 24 h later. Luciferase activity was measured with a Dual-Luciferase Reporter Assay System (Promega).

## Co-immunoprecipitation and western blotting

In brief, total protein was extracted with IP-buffer (150 mM NaCl, 1% NP-40, 10 mM Tris-Hcl and 1 mM EDTA) or cell lysis buffer (Sigma-Aldrich) containing a protease inhibitor (Sigma-Aldrich) and Phosphatase Inhibitor Cocktail Tablets. The cell lysates were subjected to SDS-PAGE, and immunoblot analysis was performed with the indicated antibodies. The data were collected by SageCapture (MiniChemi610, Beijing Sage Creation Science Co., Ltd.).

## Immunofluorescence

HEK293T cells were cultured on glass coverslips in 12-well plates and transfected with specific-plasmids. The cells were infected with SeV 19 h after transfection for 8 h. The cells were fixed in 4%

paraformaldehyde for 15 min at room temperature, rinsed with PBS three times (5 min/times), permeabilized with 0.1% Triton X-100 for 5 min at room temperature, and blocked with QuickBlock (Beyotime) for 15 min at room temperature. The cells were rinsed with PBS, and then stained with the indicated primary antibodies overnight at 4 °C, followed by incubation with fluorescent-dye-conjugated secondary antibodies. The nuclei were stained with DAPI. Images were acquired with x63 magnification (LSM880 Zeiss, Germany), analyzed by ZEN imaging software (2012 SP1,8.1), and quantified by Image J software (1.53c).

Crude mitochondria extraction and Semi-denaturing detergent agarose gel electrophoresis (SDD-AGE).

Crude mitochondria extraction was used to analyze the aggregation of MAVS by SDD-AGE[15]. In brief, cells were harvested by means of pre-chilled PBS and centrifuged at $600 \times g$ for 5 min at 4 °C to spin out clean cells. The cells were resuspended in mitochondrial isolation buffer (Beyotime Biotechnology) for 15 min, then subjected to dounce-homogenization to lyse the cells. Next, the cell homogenate was centrifuged at $600 \times g$ for 10 min at 4 °C to obtain mitochondrial suspension. The supernatant was centrifuged at $11,000 \times g$ for 10 min at 4 °C to obtain active mitochondria. Crude mitochondria (P5) were resuspended in 1×sample buffer (0.5× TBE, 10% glycerol, 2% SDS and 0.0025% bromophenol blue) and detected by means of SDD-AGE (1.5% vertical agarose gel).

### Viral infection and plaque assay
The cells were infected with the specified viruses (VSV with 0.1 MOI, HSV-1 with 10 MOI and SeV) for the indicated time points. For VSV plaque assay, the supernatants from VSV-infected cells were diluted and then added into overgrown HEK293 cells cultured on 24-well plates. One hour later, the supernatants were removed and washed with PBS twice. The cells were incubated with DMEM containing 2% methylcellulose for 24 h or 48 h, and then fixed with 4% paraformaldehyde for 15 min and stained with 1% crystal violet for 30 min. Plaques were counted and analyzed determine viral titer as Pfu/ml.

### In vivo experiments and tissue staining
Male mice of *Prmt9*[CKO] and *Prmt9*[WT] (6–8 weeks old) were infected with VSV ($1 \times 10^8$ Pfu/mouse) by intraperitoneal injection. The survival of mice was monitored every day (1–15 days), and the lungs, liver, spleen, or serum were collected for qRT-PCR analysis or plaque assays at 24 h after infection. Hematoxylin-and-eosin staining was performed to observe histologic changes of lungs[47].

### Protein expression and purification
Active N-GST-PRMT9 recombinant proteins were purified from insect cells[27], which were purchased from Sino Biological. cDNA of human PRMT9 (NM_138364.2) was inserted into baculovirus vector followed by generation of recombinant baculovirus. The expression and purification were performed according to the manufacturer's manual.

His-MAVS, His-MAVS (R41K), His-MAVS (R43K), and His-MAVS (R41K, R43K) recombinant proteins were purified from Prokaryotic cells as described previously[24]. In brief, cDNA of Human MAVS or MAVS mutants (Gene ID: 57506) were cloned into the vector pET30 (a). Subsequently, the aforementioned plasmids identified by means of DNA sequencing were transformed into BL21 (DE3) competent cells. For protein expression, the cells were incubated under optimal conditions, and induced with isopropyl-β-D-thiogalactopyranoside (IPTG) at 16 °C overnight. Cell pellets were collected via centrifugation and resuspended in homogenizing buffer. Pellets were homogenized and centrifuged to obtain the lysate supernatant. The supernatant was purified using Ni-NTA agarose (QIAGEN) according to the manufacturer's manual. The protein concentration of the final product was determined by means of BCA assays.

### In vitro methylation assay
In vitro methylation assay reactions were performed in 20 μl of methylation reaction buffer (20 mM Tris–HCl, pH 8.0, 200 mM NaCl, 0.4 mM EDTA,) including 1 μg of MAVS, 3 μg of recombinant enzymes GST-PRMT9 and 1 μg S-adenosylmethionine (SAM). The reaction mixture was incubated at 30 °C for 30 min and then identified by means of SDS-PAGE.

### In vitro radioactivity assay
Methylation reactions were carried out in 20 μl methylation buffer (0.5 M EDTA, 1 M DTT and Tris-Hcl (8.0)) containing 0.5 uCi S-[³H-Me] adenosylmethionine (NET155H, Perkin Elmer), 1 μM recombinant PRMT9 protein and 3 μM MAVS or MAVS mutants. The reaction mixture was incubated at 30 °C for 5 min, 15 min, 30 min, 50 min, or 60 min then spotted onto Whatman 3 mm cellulose filter paper discs (Fisher Scientific). 5% ice-cold TCA was used to wash the paper discs for three times to stop the reaction, followed by 20% ethanol. Liquid scintillation counting was used to determine the amounts of radioactivity.

### MAVS Aggregation and Activation in vitro
HA-MAVS, and HA-MAVS (R41K, R43K) plasmids were transfected into HEK293T cells. Crude mitochondria (P5) were isolated from HEK293T cells after transfection with 24 h. Firstly the P5 were incubated with recombinant PRMT9 proteins and SAM at 30 °C for 20 min as the methylation reaction mixture. Secondly, the RIG-I reaction system containing purified Flag-RIG-I, ubiquitin chains, ATP, $Mg^{2+}$ or 5'pppRNA. The reaction was incubated at 30 °C for 20 min as well. Finally, the methylation mixture and RIG-I system was incubated together 30 °C for 20 min. The reaction mixture was centrifuged at 11000r for 10 min, and then the pellets were added into loading buffer and analyzed by SDD-AGE as described.

### Statistical analysis
All data are presented as the mean SD. of at least three independent experiments unless otherwise stated. Log-rank (Mantel-Cox) text was used for statistical analysis of the survival curves. Scintillation counting assay statistical significance was determined using the Two-way ANOVA test. Statistical analyses were performed with GraphPad Prism 7.0 software. For other data, statistical significance was determined using the two-tailed Student's *t*-test, with a *p*-value < 0.05 considered statistically significant, and ns, not significant (*p* > 0.05).

## Data availability
All relevant data that support the study are provided in the article and Supplementary Information. Other relevant data can be obtained from the corresponding author upon reasonable request. Source data are provided with this paper.

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

## Acknowledgements

We thank Dr. Yiping Wang (Fudan University, China) for kindly providing Human GFP-PRMT (1-9) plasmids, Dr. Jian Li (Shandong University) for the plasmid pDsRED2-Mito. This work was supported by grants from the National key research and development program (2021YFC2300603 to C.G.) and grants from the National Natural Science Foundation of China (31730026, 81930039 to C.G., and 31900680, 82222027 to B.L.). This work was also supported by the National Postdoctoral Program for Innovative Talents (BX201700146 to B.L.) and Shandong Provincial Natural Science Foundation (ZR2018BC021, ZR2021YQ48, ZR2021ZD08 to B.L.).

## Author contributions

C.G. conceived the study; C.G. and B.L. designed and supervised the research; X.B. performed the research; C.S., F.L., T.C., L.Z., and Y.Z. contributed reagents and experimental advice; F.L., Y.Z., B.L. and L.Z. provided discussions; C.G. B.L. and X.B. analyzed the data and wrote the paper.

## Competing interests

The authors declare no competing interests.
