## [Peer Review File · Nature Communications]

The protein arginine methyltransferase PRMT9 attenuates MAVS activation through arginine methylation***Editorial Note:** Figures have been redacted from the Peer Review File.

REVIEWER COMMENTS

Reviewer #1 (Remarks to the Author):

In the current manuscript by Bai et al., the authors report that protein arginine methyltransferase 9 (PRMT9) negatively regulates innate antiviral immunity by catalyzing MAVS methylation at both Arg41 and Arg43. They showed that PRMT9 deficiency enhanced the innate antiviral response to RNA viruses both in vitro and in vivo and not to DNA virus infection (i.e., HSV). They further claim that methylation of MAVS by PRMT9 could inhibit MAVS aggregation in absence of any viral stimulation, underlying a putative role of PRMT9 in keeping MAVS inactive in physiological conditions. The methylation of MAVS by PRMT9 is not conclusive. The in vitro methylation assays are not properly performed. They use anti-Flag proteins that have been shown to also bring down PRMT5. This has led to many artifacts in the field namely with the retraction of Shank2 PRMT7 Elife paper recently PMID: 34292156. Thus, the present manuscript is interesting, but more conclusive experiments are required that MAVS is a direct substrate of PRMT9.

Major comments

1. The authors claim PRMT9 is associated with the mitochondria in HEK293 cells by IF and fractionation. The data with the IF needs to be quantified. PRMT5 and PRMT7 seem to also localize in the mitochondria? The mitochondria fraction protocol is absent in the methods and why not also blot with other PRMTs in Figure 1B? Thus the data with PRMT9 being exclusive is not complete.
2. PRMT9 should also be shown as in Fig S1 in relevant peritoneal macrophages and THP1 cells.
3. Figure S2. Are the siPRMT9 cells undergoing cell cycle arrest? Senescence as observed with many PRMT siRNAs? Is this specific for PRMT9, how about siRNA PRMT5, and PRMT7?
4. Where does Flag-PRMT9 localize in cells 1g, 1h? Where is it localized?
5. The data point for IFNB1 (VSV) siPRMT9 look the same as Figure 2 D PRMT9CKO. Look at blue bar and the identical pattern of triangles with both bars indicating ~2600. Can the authors confirm that the data points are indeed different?
6. Hematoxylin-and-eosin staining showed less infiltration of inflammatory cells into the lungs of Prmt9CKO mice than in the lungs of Prmt9WT mice following infection (Fig. 3g). This is not clear with h&e provided.
7. Flag PRMT9 seems to localize more mitochondria with SeV (Figure 4e) unlike Fig S1b. please explain.
8. Fig. 5c. Flag IP coimmunoprecipitates HA-MAVS. PRMT5 binds M2 flag (Nishioka and Reinberg 2003). So this experiment needs to be redone without FLAG IP.
9. Fig. 5d. in the methods the purification of the recombinant proteins needs to be detailed properly, bacteria, mammalian cells? if purchased from Sino Biologicals more detail are needed. Is the interaction occurring with hypomethylated MAVS?
10. Figure 6a-c Sap145 Rme2 methylation should be shown as a positive control.
11. The increase SDMA with WT PRMT9 in 6d not as strong as 6a (-SeV). Why?
12. Knockout of MAVS in 6e (lane 2) is not a complete ko. Explain.
13. If you blot 6d, 6e HA IP also with antiGFP to visualize PRMT9 or PRMT9(G260E) interaction what do you see?
14. As mentioned earlier Flag Ips contain PRMT5. Thus the data is not convincing that PRMT9 directly methylates MAVS it may be indirect by another PRMT (ie PRMT5). Thus to prove without doubt an invitro methylation reaction like Yang et al., 2015 figure 4A needs to be performed with a GST-PRMT9 (or HAPRMT9 in insect cells like Yang) and GST-MAVS. No flag and no HEK293 purified proteins.
15. Mutations of R41 and R43 are not very convincing in 6i. The sDMA antibody recognition should be 100% gone. This indicates that something is wrong.
16. Figure 6j is not convincing using Flag PRMT9 to methylate MAVS. Contaminating PRMT5 could be in the mix.

17. Figure S4 is not a good control to exclude PRMT5 as MEP50 its partner needs to be expressed in this assay.
18. MAVS aggregation on a non-denaturing gel is not convincing Fig. 7a-c, 7f , 7h. Panels d and e belong with figure 5a.
19. Line 229: The authors claim that PRMT7 could not promote MAVS methylation as shown in supplementary Fig. 4a. However, a new study published in Molecular Cell (PMID: 34171297) demonstrates that among all PRMTs, only PRMT7 catalyzed MAVS monomethylation and negatively controls MAVS-mediated antiviral innate immunity. How could the authors explain such contradictory data.
20. Line 366: The authors wrote the following 'Notably, we demonstrated that PRMT9 represents the first arginine methyltransferase which directly binds to MAVS and inhibits its function to limit RLR signaling activation.' To my knowledge, it is known and published that PRMT7 monomethylates MAVS at R52 and inhibits its interaction with RIG-I (PMID: 34171297). This statement should be revised, and rethought throughout the manuscript, especially in line with the new published study.
21. Related to Figure 4, the authors monitored by western the phosphorylation of TBK1 and IRF3 in Prmt9CKO and Prmt9WT macrophages after infection with SeV or stimulation with 5'ppp-RNA. Such stimulation/infection is known to induce I κ B α , IRF3 phosphorylation and IRF3 dimerization. Please show the immunoblots related to I κ B α (total and phosphorylated forms) following infection.
22. The authors generated a PRMT9 Crispr-Cas9 Raw264.7 cell lines (murine macrophages). However, in almost all their experiments they did not take advantage of it. They performed all their exp in vitro in HEK 293T cells. The authors need to show co-IP experiments (PRMT9-MAVS) as well as the arginine methylation of MAVS in Raw264.7 cell line.
23. It is known that RIG-I interacts with MAVS and activates the antiviral signaling pathway. Does arginine methylation of MAVS at R41 and R43 by PRMT9 alter RIG-I interaction?
24. Figure 2d: Immunoblot PRMT9 is missing
25. Figure 3b: Why are we still seeing the PRMT9 expression in PRMT9CKO peritoneal macrophage cells?
26. Statistical analyses are missing for Fig. 5a, Fig. 6j and Fig. 7d and 7g
27. Please Add the molecular weight for all immunoblots in the entire manuscript.

Minor edits:

- line 42 typo
- Line 85: 'PRMT9 mainly colocalized with mitochondria with the mitochondria' REPETITION
- Line 100 and 110: Try to be consistent with the gene/mRNA name (small/capital letters)
- Line 162 english
- Line 213: Replace 'PRMR9' by PRMT9
- Line 308: 'from' repeated two times
- Line 317: our data not 'date'
- Line 749: Replace 'miffle' by middle

Reviewer #2 (Remarks to the Author):

Summary:

In this manuscript, the authors identify the protein methyltransferase PRMT9 as a novel regulator of MAVS activation. Using PRMT9 ablation experiments (siRNA/ KO), the authors demonstrate in cell lines as well as primary macrophages that PRMT9 loss increases IFN β production upon RNA virus infections. Consequently, viral titers were demonstrated to be lower in the PRMT9-deficient cells. In line with these results, PRMT9 KO mice phenocopied the increase in RNA-virus induced cytokines, as well as a decrease in viral titers and death.

Cell-based and in vitro experiments identified that PRMT9 interacts with MAVS, explaining why PRMT9 KO affected RNA-virus induced cytokines, but not HSV-induced cytokines. Exogenous expression studies in HEK cells identified that PRMT9 catalytic activity is required for MAVS inhibition, and that PRMT9 methylates R41/R43 in MAVS in cells and in vitro. Last, the authors demonstrate that

PRMT9 expression inhibits WT MAVS aggregation on mitochondria, which is required for its activation. In contrast, a MAVS(R41/43K) mutant was no longer methylated by PRMT9, and was longer aggregation prone. Overall, the authors present a novel role for PRMT9 methyl-transferase activity, acting on MAVS to inhibit its activation.

Overall reception:

To my knowledge, no studies have been previously published describing a role of PRMT9 regulation of MAVS. In my opinion, the presented work is novel, and describes a cellular process of interest to a wider audience. Overall, the presented results are convincing and span a broad range of experiments in cell lines, primary cells, mice, and reconstituted in vitro assays. While the presented work raises many questions for follow-up work, in my opinion the conclusions by the authors are supported by the presented data, and present a substantial, new body of work. I do not have any comments which would need to be addressed with new experiments.

Main comments, which in my opinion need to be addressed in a revised manuscript:

- 1) The claimed differences in immune cell infiltrates and inflammation in the knockout mouse lungs (Fig. 3g) would be more convincing if combined with a blinded pathology score, or another quantifiable measure. It would also be of interest if the authors could add a short statement whether under mock-infection conditions there is a difference in the knock-out mice.
- 2) Fig. 6f: in vitro assay according to legend, but it says 'WCL' in the labeling. Please clarify.
- 3) The descriptions of the in vitro assays in the figure legend of Fig. 6f/j would benefit from 1-2 extra sentences describing in brief how the experiments were performed.
- 4) Supl. Fig 1a: The loss of PRMT9 at the mitochondria is not very convincing; it rather seems like an overall drop in GFP intensity. Quantification with a cross-section tool for green-red co-occurrence should be included (at least for PRMT9)
- 5) Supl. Fig. 6: the left part of the drawing seems to imply that PRMT9 links MAVS to the mitochondria in a methyl-dependent manner. It would imo be better to draw MAVS directly associated with the mitochondria as on the right side, and implement PRMT9 such that it makes clear that its methyl-transferase activity prevents MAVS aggregation.
- 6) While the manuscript is easy to read and understand, its scientific English (grammar, spelling, consistency) should be improved by a native speaker before publication. Especially the introduction and first part of the results will need some work.
- 7) All protein blots should have molecular weight markers indicated; essential basic requirement for any reader to assess whether presented bands match the predicted MW of the detected protein.

Reviewer #3 (Remarks to the Author):

Bai et al. showed that PRMT9 catalyzes methylation of MAVS. Upon virus infection, PRMT9 dissociates from mitochondria, which leads to aggregation and activation of MAVS. Many careful in vitro and in vivo experiments were performed clearly showing enhanced antiviral responses upon deletion of PRMT9.

Why only conditional PRMT9 CKO with a macrophage-selective deletion were studied. Data with complete knockout should be included.

Conditional PRMT9 CKO mice were studied in context of HSV-1 infection as a control. However, recent studies showed a role of MAVS also in HSV-1 infection. Thus, the mice should be studied more thoroughly in order not to miss important phenotypes.

Already in the abstract, the authors highlighted that "spontaneous aggregation of MAVS can lead to autoimmune diseases". However, this correlation is not so clear, yet. For example in case of SLE it was shown that indeed some patients showed MAVS aggregation, but not all. The authors should more thoroughly discuss the current knowledge about MAVS aggregation and autoimmunity in the introduction. They actually did not cite a single paper on this matter.

Immunoblot studies showed representative data of one experiment. It should be considered to quantify bands and to show data of more than one experiment. In these experiments, it should be

considered to include also PRMT9 inhibition studies e.g. by miRNA.

The authors did not mention that PRMT9 is also known as FBXO11 and VIT1. Accordingly, they missed highly relevant literature. Most importantly, in the human population more than 20 PRMT9 variants have been identified, some of which are associated with a syndromic form of intellectual disability with behavioral problems and dysmorphisms. This should be discussed and also associations with autoimmunity should be addressed.

The authors should cite more original publications and less reviews, at least for the highly relevant aspects of this study.

The manuscript is very difficult to read due to far too many typos and language insufficiencies.

Thank you very much for your questions, comments and suggestions. We have made extensive and careful revisions, with the revised parts marked in red color in our manuscript. These very insightful questions, comments and suggestions have significantly improved the manuscript and we hope that our reply has fully addressed your concerns.

The point-by-point answers to the questions, comments and suggestions were listed as below.

Reviewer #1 (Remarks to the Author):

In the current manuscript by Bai et al., the authors report that protein arginine methyltransferase 9 (PRMT9) negatively regulates innate antiviral immunity by catalyzing MAVS methylation at both Arg41 and Arg43. They showed that PRMT9 deficiency enhanced the innate antiviral response to RNA viruses both in vitro and in vivo and not to DNA virus infection (i.e., HSV). They further claim that methylation of MAVS by PRMT9 could inhibit MAVS aggregation in absence of any viral stimulation, underlying a putative role of PRMT9 in keeping MAVS inactive in physiological conditions. The methylation of MAVS by PRMT9 is not conclusive. The in vitro methylation assays are not properly performed. They use anti-Flag proteins that have been shown to also bring down PRMT5. This has led to many artifacts in the field namely with the retraction of Shank2 PRMT7 Elife paper recently PMID: 34292156. Thus, the present manuscript is interesting, but more conclusive experiments are required that MAVS is a direct substrate of PRMT9.

Answer: Thank you very much for your constructive comments and suggestions. To address your concerns, we have properly conducted new experiments or repeated some experiments in order to confirm the function of PRMT9 on MAVS methylation in the revised manuscript.

1. The authors claim PRMT9 is associated with the mitochondria in HEK293 cells by

IF and fractionation. The data with the IF needs to be quantified. PRMT5 and PRMT7 seem to also localize in the mitochondria? The mitochondria fraction protocol is absent in the methods and why not also blot with other PRMTs in Figure 1B? Thus the data with PRMT9 being exclusive is not complete.

Answer: We appreciate your constructive comments. To answer your question, Image J software was used to quantitatively analyze the colocalization between PRMT1-9 with mitochondria in HEK293T cells. Cells expressing both PRMTs and DsRED2-Mitored were selected randomly for colocalization analysis. Pearson's Coefficient was quantified by Scatter J, and drawn by GraphPad Prism 7.0. As shown in Fig S1 a - c, PRMT1, PRMT2, PRMT3, PRMT4, PRMT5, PRMT6 and PRMT8 failed to colocalize with the mitochondria regardless of infection with SeV, whereas PRMT7 colocalized with mitochondria in HEK293T cells and the colocalization was not affected by SeV infection. Notably, PRMT9 colocalized with the mitochondria and dissociated from the mitochondria upon SeV infection in HEK293T cells.

According to your suggestion, we have supplemented the protocol of mitochondria extraction in the methods sections of the main text. We have extracted the crude mitochondria from peritoneal macrophages (PM) or THP1 cells and performed western blot analysis of the protein expression of PRMT1-9 in total cell lysates or crude mitochondria lysates infected with SeV for 12 h or uninfected. As shown in Fig. S2, the immunoblot results are consistent with the confocal microscope images, indicating the specificity of PRMT9 localization in mitochondria. We added the new data in Figure S1 and S2 to the revised manuscript.

Two figures have been redacted here.

2. PRMT9 should also be shown as in Fig S1 in relevant peritoneal macrophages and THP1 cells.

Answer: We accept the suggestions from this reviewer. We detected the colocalization of the endogenous PRMT9 with mitochondria in peritoneal macrophages or THP1. PRMT9-TOMM20 colocalization was quantified using Pearson's correlation coefficient method, and immunoblot was quantified by Image J software (PRMTs were normalized with individual actin or tomm20). Confocal microscope imaging (Fig. S2 e, f) and western blot (Fig. S2a - d) data revealed that PRMT9 colocalized with mitochondria and dissociated from the mitochondria after infection with SeV. We added these new data in Figure S2 in the revised manuscript.

One figure has been redacted here.

3. Figure S2. Are the siPRMT9 cells undergoing cell cycle arrest? Senescence as observed with many PRMT siRNAs? Is this specific for PRMT9, how about siRNA PRMT5, and PRMT7?

Answer: Thank you for pointing out these interesting questions. According to your comments, we performed propidium iodide (PI) staining and f3-galactosidase (f3-gal) staining assay in PMs or THP1 cells to test the impact of PRMT9 on cell cycle

progression and senescence. The data showed that PRMT9 did not have significant effect on the cell cycle progression and senescence of PM and THP1 cells.

Currently, studies showed that loss of type I Protein Arginine Methyltransferase (PRMT3, PRMT4, PRMT5, PRMT6, and PRMT8) activity inhibited the growth of various MLL leukemia cell lines[1, 2]. As reported, the studies on cell cycle arrest and senescence for PRMTs have mainly focused on tumor cells, but findings for PRMT5 and PRMT7 in regard to cell cycle and senescence in PMs or THP1 remain unclear.

One figure has been redacted here.

4. Where does Flag-PRMT9 localize in cells 1g, 1h? Where is it localized?

Answer: To answer your question, we have tested the colocalization between Flag-PRMT9 with DsRED2-Mitored in HEK293T cells. The positive cells that expressed both PRMTs and DsRED2-Mitored were selected randomly for Pearson's correlation coefficient analysis. The data showed that Flag-PRMT9 was mostly localized with

DsRED2-Mitored in HEK293T cells. Consistent with GFP-PRMT9, Flag-PRMT9 dissociated from the mitochondria after VSV infection.

One figure has been redacted here.

5. The data point for IFNB1 (VSV) siPRMT9 look the same as Figure 2 D PRMT9CKO. Look at blue bar and the identical pattern of triangles with both bars indicating ~2600. Can the authors confirm that the data points are indeed different.

Answer: We thank the reviewer for the critical reading of our manuscript. Although the bar chart of Fig. 2d and Fig. S2h are looking similar, the data are different. The original data are shown below (Fig. 2d and Fig. S2h).

To avoid any misunderstandings, we repeated the experiment in Fig. S2h in THP1 cells, and updated these new data in Fig. S2h. Fig. S2 was renamed as Fig. S3 in the revised manuscript.

Two figures have been redacted here.

6. Hematoxylin-and-eosin staining showed less infiltration of inflammatory cells into the lungs of Prmt9CKO mice than in the lungs of Prmt9WT mice following infection (Fig. 3g). This is not clear with h&e provided.

Answer: In response to this suggestion, we provided a new clear Hematoxylin-and-eosin (H&E) staining (Fig. 3g). The H&E analysis showed that PRMT9-deficient alleviated inflammatory cell infiltration, tissue edema and pulmonary fibrosis. In addition, we provided a quantification histogram of the infiltration of immune cells (Fig. 3g). Original H&E staining was replaced with the new one in the revised manuscript.

One figure has been redacted here.

7. Flag PRMT9 seems to localize more mitochondria with SeV (Figure 4e) unlike Fig S1b. please explain.

Answer: We thank the reviewer for the critical reading of our manuscript. In Fig. 4e, the co-localization of PRMT9 and mitochondria was not detected, so the confocal microscope imaging data only showed the expression of PRMT9 in the cytoplasm. We optimized the experimental conditions and repeated the experiment in HEK293T cells. The new immunofluorescence images were replaced the old ones in the revised manuscript. Fig. 4e was renamed as Fig. 4i in the revised manuscript.

One figure has been redacted here.

8. Fig. 5c. Flag IP coimmunoprecipitates HA-MAVS. PRMT5 binds M2 flag (Nishioka

and Reinberg 2003). So this experiment needs to be redone without FLAG IP.

Answer: We appreciate the constructive comments from this reviewer. We repeated the Co-IP using GFP-PRMT9 plasmids to investigate their interactions. Co-IP and western blot analysis showed that PRMT9 interacted with MAVS but not with cGAS, RIG-I, MDA5, TBKI, and IRF3 (Fig. 5c). The previous blots were replaced with the new blots in the revised manuscript

One figure has been redacted here.

9. Fig. 5d. in the methods the purification of the recombinant proteins needs to be detailed properly, bacteria, mammalian cells? if purchased from Sino Biologicals more detail are needed. Is the interaction occurring with hypomethylated MAVS?

Answer: The reviewer's valuable suggestions were critical for the improvement of our manuscript. We have added detailed instructions regarding the purification of PRMT9 and MAVS protein in the revised manuscript.

As shown in Fig. 6d, the PRMT9 enzymatic mutant G260E lost the ability to induce MAVS methylation, but the interaction between PRMT9 G260E and MAVS was not affected. The results indicate that the methylation of MAVS had no effect on the interaction with PRMT9.

One figure has been redacted here.

0. Figure 6a-c Sap145 Rme2 methylation should be shown as a positive control.

Answer: We appreciate the valuable suggestions from the reviewer. It has been reported that PRMT9 is a type II PRMT that methylates SAP145 [3]. To definitively confirm the

methylation function of PRMT9 on MAVS, we added Sap145 as the positive control. As shown in Fig. 6a and c, overexpression of PRMT9 could promote the methylation of MAVS as well as SAP145, while PRMT9-deficiency alleviated the methylation of both MAVS and SAP145. Overall, the data demonstrates that PRMT9 promoted MAVS arginine methylation. We added the new data to the revised manuscript

One figure has been redacted here.

11. The increase SDMA with WT PRMT9 in 6d not as strong as 6a (-SeV). Why?

Answer: We'd like to thank the reviewer for the critical reading of our manuscript. The slight differences in SDMA intensity may be influenced by multiple factors, such as the use of different batches of cells, differences in cell state, and differences in plasmid expression activity. In order to answer the questions, we optimized the experimental conditions and repeated this experiment. The new data were added in the revised manuscript.

One figure has been redacted here.

12. *Knockout of MAVS in 6e (lane 2) is not a complete ko. Explain.*

Answer: Yes, we agree with the reviewer. Primary peritoneal macrophages in lane 2 were prepared from *Prmt9^{loxp/loxp}* mice with overexpression of Lyz2-Cre. The Lyz2-Cre sometimes is not working very efficiently that have reported in some literatures [4-7]. Therefore, the PRMT9 in lane 2 could still be detected, but much lower than lane 1.

13. *If you blot 6d, 6e HA IP also with anti-GFP to visualize PRMT9 or PRMT9(G260E) interaction what do you see?*

Answer: We appreciate the valuable suggestions. We performed Co-IP and western blot to test the interaction between GFP-PRMT9 (or endogenous PRMT9) and HA-MAVS (or endogenous MAVS). As shown in Fig. 6d and 6e, MAVS could bind to both PRMT9(WT) and PRMT9(G260E). Thus, the data demonstrated that PRMT9 catalyzed MAVS methylation through the methyltransferase activity thereof rather than PRMT9/MAVS interaction.

One figure has been redacted here.

14. *As mentioned earlier Flag Ips contain PRMT5. Thus the data is not convincing PRMT9 directly methylates MAVS it may be indirect by another PRMT (ie PRMT5). Thus to prove without doubt an invitro methylation reaction like Yang et al., 2015 figure 4A needs to be performed with a GST-PRMT9 (or HAPRMT9 in insect cells like Yang) and GST-MAVS. No flag and no HEK293 purified proteins.*

Answer: We thank the reviewer raising this critical question and fully agree with the reviewer. In order to further strengthen our study, we have purified GST-PRMT9 from insect cells [3], and His-MAVS recombinant proteins from E.coli for the *in vitro* methylation and interaction assay. We have repeated the experiments in Fig. 5d and Fig. 6f using the new purified proteins. The results are consistent with our previous results. These results demonstrate that PRMT9 methylates MAVS directly.

Two figures have been redacted here.

15. Mutations of R41 and R43 are not very convincing in 6i. The sDMA antibody recognition should be 100% gone. This indicates that something is wrong.

Answer: We thank the reviewer and agree with the reviewer's comments that the SDMA antibody recognition should be 100% gone in Fig. 6i. In the previous experiments, MAVS recombinant proteins were purified from HEK293 cells, so it may be contaminated with endogenous MAVS or PRMT5[3]. To further prove our conclusion, we have purified PRMT9 from insect cells, and MAVS from E. coli to perform the experiments. The new data are shown in Fig. 6i.

One figure has been redacted here.

16. Figure 6j is not convincing using Flag PRMT9 to methylate MAVS. Contaminating PRMT5 could be in the mix.

Answer: We are very grateful for the reviewer's comments and have repeated the experiment using the new purified proteins. GST-PRMT9 proteins was purified from insect cells[3], whereas His-MAVS, His-MAVS(R41K), His-MAVS(R43K), and His-MAVS(R41/43K) proteins were purified from E. coli. We performed an in vitro methylation assay with S-[3H-Met] adenosylmethionine as the methyl donor. Fig. 6j showed that the radioactivity of His-MAVS was greatly increased in the presence of recombinant GST-PRMT9 protein, whereas the radioactivity of MAVS (R41K), MAVS (R43K) and MAVS (R41K, R43K) was significantly decreased compared to that of MAVS (WT). The new data were added in the revised manuscript.

One figure has been redacted here.

17. Figure S4 is not a good control to exclude PRMT5 as MEP50 its partner needs to be expressed in this assay.

Answer: This suggestion was greatly appreciated. We optimized the experimental conditions and repeated this experiment using MEP50, PRMT5, PRMT7, and PRMT9 overexpressing plasmids in HEK293T. As shown in Fig. S4, PRMT5 and MEP50 together could not promote MAVS methylation.

One figure has been redacted here.

0. MAVS aggregation on a non-denaturing gel is not convincing Fig. 7a-c, 7f, 7h. Panels d and e belong with figure 5a.

Answer: We accept the suggestion from the reviewer. We have optimized the experimental conditions and try to provide more convincing blots. The MAVS aggregation showed in Fig. 7a-c, 7f, 7h were repeated in more than three independent experiments. These results consistent with the previous experimental results.

One figure has been redacted here.

19. Line 229: The authors claim that PRMT7 could not promote MAVS methylation as shown in supplementary Fig. 4a. However, a new study published in Molecular Cell (PMID: 34171297) demonstrates that among all PRMTs, only PRMT7 catalyzed MAVS monomethylation and negatively controls MAVS-mediated antiviral innate immunity. How could the authors explain such contradictory data.

Answer: We are very grateful for the comments from this reviewer. During the process of submission of our manuscript, the Molecular Cell paper is not published, therefore we did not add the PRMT7 in our manuscript. To better answer the reviewer's question, we used Image J to quantitatively analyze the immunoblotting of the previous data in Fig. S4. The result showed that PRMT7 could slightly increase the level of MAVS MMA (0.16 to 0.28), but not SDMA (0.19 to 0.21), which is consistent with results in the Molecular Cell paper. These data are also consistent with our results in Figure S1 and S2, in which PRMT7 colocalize with mitochondria. However, compared to PRMT9, PRMT7 still localized on mitochondria after virus infection, which indicate PRMT7 may inhibit MAVS activation constitutively. While, PRMT9 may represent a more physiological inhibitor for MAVS, because the viral infection could induce the

dissociation of MAVS from mitochondria.

In order to definitively confirm the methylation function of PRMT7 on MAVS, we have optimized the experimental conditions and repeated this experiment. Consistent with the previous experimental results, PRMT7 slightly increased the MMA of MAVS (0.10 to 0.24), but not SDMA (0.13 to 0.11). Besides, the results showed that only PRMT9 efficiently promoted symmetrical arginine dimethylation (SDMA) of MAVS. We have rewritten this part in the revised manuscript.

One figure has been redacted here.

20. Line 366: The authors wrote the following ‘Notably, we demonstrated that PRMT9 represents the first arginine methyltransferase which directly binds to MAVS and inhibits its function to limit RLR signaling activation.’ To my knowledge, it is known and published that PRMT7 monomethylates MAVS at R52 and inhibits its interaction with RIG-I (PMID: 34171297). This statement should be revised, and rethought throughout the manuscript, especially in line with the new published study.

Answer: We agree with the reviewer and have rewritten this part in the revised manuscript. Since PRMT7 (PMID: 34171297) had not been published when we submitted the manuscript to Nature Communications, therefore, we did not cite the PRMT7 work.

21. Related to Figure 4, the authors monitored by western the phosphorylation of TBK1

and IRF3 in Prmt9CKO and Prmt9WT macrophages after infection with SeV or stimulation with 5'ppp-RNA. Such stimulation/infection is known to induce IκBα, IRF3 phosphorylation and IRF3 dimerization. Please show the immunoblots related to IκBα (total and phosphorylated forms) following infection.

Answer: We thank the reviewer for this valuable suggestion. As shown in Fig. 4, we repeated this experiment and examined the expression of IκBα (total and phosphorylated forms). In addition, Image J software was used to quantitatively analyze the immunoblotting seen in Fig. 4a-f of three independent repeated experiments. Ratio: p-TBK1/TBK1, p-IRF3/IRF3, or p-IκBα/IκBα. These new data were added to the revised manuscript.

One figure has been redacted here.

22. The authors generated a PRMT9 Crispr-Cas9 Raw264.7 cell lines (murine macrophages). However, in almost all their experiments they did not take advantage of it.

They performed all their exp in vitro in HEK 293T cells. The authors need to show co-IP experiments (PRMT9-MAVS) as well as the arginine methylation of MAVS in Raw264.7 cell line.

Answer: We accept the suggestion from the reviewer. And, we have performed the Co-IP experiments and MAVS methylation experiments in Raw264.7 cell line (Fig. S7a, b).

One figure has been redacted here.

23. It is known that RIG-I interacts with MAVS and activates the antiviral signaling pathway. Does arginine methylation of MAVS at R41 and R43 by PRMT9 alters RIG-I interaction?

Answer: We thank the reviewer for the kind suggestion. As suggested, we have performed a new experiment using WT MAVS and MAVS mutant R41/43K to check whether PRMT9-mediated MAVS-R41/43 methylation affect the interaction between RIG-I and MAVS. The data showed that PRMT9-mediated MAVS arginine methylation did not regulate MAVS binding to RIG-I.

One figure has been redacted here.

24. Figure 2d: Immunoblot PRMT9 is missing

Answer: We thank the reviewer for the critical reading. We have added the expression of PRMT9 in the immunoblot, which is now revised in Fig. 2d.

One figure has been redacted here.

15. *Figure 3b. Why we are still seeing the PRMT9 expression in PRMT9CKO peritoneal macrophage cells?*

Answer: We thank the reviewer for raising these concerns. Similar to question 12, we used Lyz2-Cre to delete PRMT9 expression in macrophages. The Lyz2-Cre sometimes is not working very efficiently that have reported in some literatures [4-7]. Therefore, the PRMT9 could still be detected, but much lower than that in *Prmt9^{loxp/loxp}* without Lyz2-Cre.

16. *Statistical analyses are missing for Fig. 5a, Fig. 6j and Fig. 7d and 7g*

Answer: We apologize for not including the statistical analysis in Fig. 5a, Fig. 6j and Fig. 7d and 7g. Statistical significance was determined by a two-tailed Student's t-test in Fig. 5a, Fig. 7d, 7g; and Fig. 6j. The relevant content has been corrected in the revised manuscript.

17. *Please Add the molecular weight for all immunoblots in the entire manuscript.*

Answer: We thank the reviewer for these suggestions. We have added molecular weight in all immunoblots in all figures.

Minor edits

-line 42 typo

-Line 85. 'PRMT9 mainly colocalized with mitochondria with the mitochondria'

REPETITION

-Line 100 and 110. Try to be consistent with the gene/mRNA name (small/capital letters)

-Line 162 english

-Line 213. Replace 'PRMR9' by PRMT9

-Line 308. 'from' repeated two times

-Line 317. our data not 'date'

-Line 749. Replace 'miffle' by middle

Answer: We thank the reviewer for the critical reading of our manuscript. We have corrected typos and carefully checked the whole manuscript to avoid such mistakes.

Reviewer #2 (Remarks to the Author):

Summary:

In this manuscript, the authors identify the protein methyltransferase PRMT9 as a novel regulator of MAVS activation. Using PRMT9 ablation experiments (siRNA/ KO), the authors demonstrate in cell lines as well as primary macrophages that PRMT9 loss increases IFN β production upon RNA virus infections. Consequently, viral titers were demonstrated to be lower in the PRMT9-deficient cells. In line with these results, PRMT9 KO mice phenocopied the increase in RNA-virus induced cytokines, as well as a decrease in viral titers and death.

Cell-based and in vitro experiments identified that PRMT9 interacts with MAVS, explaining why PRMT9 KO affected RNA-virus induced cytokines, but not HSV-induced cytokines. Exogenous expression studies in HEK cells identified that PRMT9 catalytic activity is required for MAVS inhibition, and that PRMT9 methylates R41/R43 in MAVS in cells and in vitro. Last, the authors demonstrate that PRMT9 expression inhibits MAVS aggregation on mitochondria, which is required for its activation. In contrast, MAVS(R41/43K) mutant was no longer methylated by PRMT9, and was longer aggregation prone. Overall, the authors present a novel role for PRMT9 methyltransferase activity, acting on MAVS to inhibit its activation.

Overall reception:

To my knowledge, no studies have been previously published describing a role of PRMT9 regulation of MAVS. In my opinion, the presented work is novel, and describes a cellular process of interest to a wider audience. Overall, the presented results are convincing and span a broad range of experiments in cell lines, primary cells, mice, and reconstituted in vitro assays. While the presented work raises many questions for follow-up work, in my opinion the conclusions by the authors are supported by the presented data, and present a substantial, new body of work. I do not have any comments which would need to be addressed with new experiments.

Answer: We thank the reviewer for recognizing the novelty and significance of our study. We also thank the reviewer for the excellent advice in helping to improve our manuscript.

Main comments, which in my opinion need to be addressed in a revised manuscript:

1) The claimed differences in immune cell infiltrates and inflammation in the knockout mouse lungs (Fig. 3g) would be more convincing if combined with a blinded pathology score, or another quantifiable measure. It would also be of interest if the authors could add a short statement whether under mock-infection conditions there is a difference in the knock-out mice.

Answer: We thank the reviewer for the insightful suggestions. Lung tissue sections of mice after VSV infection were stained with hematoxylin and eosin (H&E). Inflammation scores of lung tissue were quantitated by ImageJ and visualized with GraphPad Prism 7. The H&E analysis showed that PRMT9-deficient alleviated inflammatory cell infiltration, tissue edema and pulmonary fibrosis (Fig. 3g). Also, we

added a statement to indicate that there is no significant difference in the lung between WT and CKO mice without VSV infection in our revised manuscript.

One figure has been redacted here.

2) *Fig. 6f: in vitro assay according to legend, but it says 'WCL' in the labeling. Please clarify.*

Answer: We thank the reviewer for critical reading of our manuscript. It should be marked 'input' instead of 'WCL'. We have corrected it and carefully checked the whole manuscript to rectify additional mistakes.

3) *The descriptions of the in vitro assays in the figure legend of Fig. 6f/j would benefit from 1-2 extra sentences describing in brief how the experiments were performed.*

Answer: We are grateful to the reviewer's suggestions, and have rewritten this part in the revised figure legend.

0) *Supl. Fig 1a: The loss of PRMT9 at the mitochondria is not very convincing; it rather seems like an overall drop in GFP intensity. Quantification with a cross-section tool for green-red co-occurrence should be included (at least for PRMT9)*

Answer: We agree with the reviewer. We have repeated this experiment by expressing GFP-PRMT9 and DsRED2-Mitored in HEK293T cells to confirm the colocalization between PRMT9 with mitochondria. In addition, Pearson's correlation coefficient method was used to quantify PRMT9-mitochondria co-localization to further confirm that PRMT9 colocalized with mitochondria, and dissociated from the mitochondria upon SeV infection in HEK293T cells (Fig. S1).

One figure has been redacted here.

5) Supl. Fig. 6: the left part of the drawing seems to imply that PRMT9 links MAVS to the mitochondria in a methyl-dependent manner. It would imo be better to draw MAVS directly associated with the mitochondria as on the right side, and implement PRMT9 such that it makes clear that its methyl-transferase activity prevents MAVS aggregation.

Answer: We thank the reviewer for the constructive comments. As suggested, we have redrawn the schematic model to show clearly the role of methyl-transferase activity of PRMT9 to prevent MAVS aggregation. Fig. S6 was renamed as Fig. S9 in the revised manuscript.

One figure has been redacted here.

6) While the manuscript is easy to read and understand, its scientific English (grammar, spelling, consistency) should be improved by a native speaker before publication. Especially the introduction and first part of the results will need some work.

Answer: We thank the reviewer for the critical reading of our manuscript, and we have corrected the mistakes and polished the whole manuscript.

1) All protein blots should have molecular weight markers indicated; essential basic requirement for any reader to assess whether presented bands match the predicted MW of the detected protein.

Answer: We accept the suggestions from the reviewer and have added the molecular weight for all immunoblots in all figures.

Reviewer #3 (Remarks to the Author):

Bai et al. showed that PRMT9 catalyzes methylation of MAVS. Upon virus infection, PRMT9 dissociates from mitochondria, which leads to aggregation and activation of MAVS. Many careful in vitro and in vivo experiments were performed clearly showing enhanced antiviral responses upon deletion of PRMT9.

Why only conditional PRMT9 CKO with a macrophage-selective deletion were studied. Data with complete knockout should be included.

Conditional PRMT9 CKO mice were studied in context of HSV-1 infection as a control.

However, recent studies showed a role of MAVS also in HSV-1 infection. Thus, the mice should be studied more thoroughly in order not to miss important phenotypes.

Answer: We thank the reviewer for summarizing the key results of our work, and raising this crucial question. To address the reviewer's concerns, we have performed the following experiments:

1) In order to establish complete knockout mice, *Prmt9^{loxp/loxp}* mice were mated with Cre-ERT2 mice to induce PRMT9 systemic knockout mice as reported [8, 9]. As suggested, to further confirm the function of PRMT9, *Cre^{ERT} Prmt9^{fl/fl}* mice (call *Prmt9^o* here) were infected with VSV (1.8×10^7 PFU per mouse) for 24 h by tail vein injection (n = 6 mice per group). After the mice were sacrificed, the serum samples, lung, liver, and spleen were collected for the experiments. Consistent with Fig. 3 in macrophage specific deletion of PRMT9, PRMT9 complete deficiency enhanced the antiviral innate immune response in vivo against RNA virus (Fig. S8 a - d).

2) *Prmt9^o* and *Prmt9^{WT}* mice were injected with HSV-1 (1.5×10^8 PFU per mouse) for 24 h by tail vein injection (n = 6 mice per group). After the mice were sacrificed, the serum samples were tested for the secretion of IFN- β protein, and the brains were used to observe the *Ifnb1* mRNA, the copy number of HSV-1 genomic DNA and viral titer of HSV-1. As shown in Fig. S8e and f, the production of IFN- β protein and *Ifnb1* mRNA was barely affected by *Prmt9*-knockout. Consistent with the expression with *Ifnb1*, the results showed that the copy number of HSV-1 genomic DNA and viral titer of HSV-1 did not produce an effect in *Prmt9^o* mice relative to that in *Prmt9^{WT}* mice.

3) Consistently, survival of *Prmt9^o* and *Prmt9^{WT}* mice (n=12 mice per group, 6-8 weeks old) after tail vein injection with VSV (1×10^8 PFU per mouse) or HSV-1 (1.5

× 10⁸ PFU per mouse), we found that *Prmt9*^O mice were less susceptible than *Prmt9*^{WT} mice to infection with VSV (Fig. S8g). But for infection with HSV-1, there were no differences observed in the survival of *Prmt9*^O mice and *Prmt9*^{WT} mice (Fig. S8h).

All together, similar to macrophage specific deletion mice, PRMT9 complete deficiency enhanced the antiviral innate immune response *in vivo* against RNA virus, but not for HSV-1.

One figure has been redacted here.

Already in the abstract, the authors highlighted that “spontaneous aggregation of MAVS can lead to autoimmune diseases”. However, this correlation is not so clear, yet. For example in case of SLE it was shown that indeed some patients showed MAVS aggregation, but not all. The authors should more thoroughly discuss the current knowledge about MAVS aggregation and autoimmunity in the introduction. They actually did not cite a single paper on this matter.

Answer: We accept the insightful suggestions from the reviewer. As suggested, the relevant content has been rewritten in the revised manuscript. And literatures to support this claim have been added.

Immunoblot studies showed representative data of one experiment. It should be considered to quantify bands and to show data of more than one experiment. In these experiments, it should be considered to include also PRMT9 inhibition studies e.g. by miRNA

Answer: We thank the reviewer for the valuable suggestions. Image J software was used to quantitatively analyze most of the immunoblot data in the article. Histograms for the quantitative analysis of the three independent repeated experiments were added in revised manuscript.

According to your suggestion, we conducted a new experiment in order to study PRMT9 inhibition by microRNA. As reported[10], miRNA-543 directly binds to 3'-UTR of PRMT9 mRNA to inhibit PRMT9 translation. Indeed, overexpression of miR-543 significantly decreased PRMT9 expression. Consistently, overexpression of miR-543 significantly increased the *Ifnb1*, *Ifna4*, and *Cxcl10* mRNA level after infection with SeV, VSV or stimulation with 5'PPP RNA compared to the NC group in macrophages.

One figure has been redacted here.

The authors did not mention that PRMT9 is also known as FBXO11 and VIT1. Accordingly, they missed highly relevant literature. Most importantly, in the human population more than 20 PRMT9 variants have been identified, some of which are associated with a syndromic form of intellectual disability with behavioral problems and dysmorphisms. This should be discussed and also associations with autoimmunity should be addressed.

Answer: We thank the reviewer for the valuable suggestions. In the past, the catalytic activity of PRMT9 was unknown and named FBXO11 until Yang et al identified it as the second SDMA-forming enzyme in mammals [3, 11]. Several studies have shown that FBXO11 was previously erroneously named PRMT9[3, 11, 12]. A recent study demonstrated that PRMT9 has two distinguishing features: AdoMet-binding motif, and three N-terminal tetratricopeptide repeats (TPR), a hallmark amino acid sequence motif belonging to the PRMT family. Studies have proved that FBXO11 does not contain the

PRMT family distinguishing features. In addition, the nematode homolog of FBXO11 has been observed no methyltransferase activity [13]. Therefore, researchers believe that the arginine methyltransferase activity of FBXO11 was most likely the result of contamination of the FLAG-tagged protein with PRMT5[14, 15]. To avoid confusion, FBXO11 was not described in our manuscript. According to your suggestion, we queried the literature that studied on PRMT9, and we found two PRMT9 variants, none of which have been reported the associations between PRMT9 variants and autoimmunity. We also found that the current studies about mild-moderate intellectual disability and some autoimmune diseases were associated with FBXO11 variants[16].

The authors should cite more original publications and less reviews, at least for the highly relevant aspects of this study.

Answer: We are extremely grateful for the reviewer's comments and have corrected this in the revised manuscript.

The manuscript is very difficult to read due to far too many typos and language insufficiencies.

Answer: We apologize for the mistakes in manuscript. We have corrected the mistakes and polished the whole manuscript.

References:

1. Wang, C., et al., *Development of Potent Type I Protein Arginine Methyltransferase (PRMT) Inhibitors of Leukemia Cell Proliferation*. J Med Chem, 2017. **60**(21): p. 8888-8905.
2. Phalke, S., et al., *p53-Independent regulation of p21Waf1/Cip1 expression and senescence by PRMT6*. Nucleic Acids Res, 2012. **40**(19): p. 9534-42.
3. Yang, Y., et al., *PRMT9 is a type II methyltransferase that methylates the splicing factor SAP145*. Nat Commun, 2015. **6**: p. 6428.
4. Zhao, L., et al., *Generation and identification of a conditional knockout allele for the PSMD11 gene in mice*. BMC Dev Biol, 2021. **21**(1): p. 4.
5. Li, S., et al., *The mitochondrial protein ERAL1 suppresses RNA virus infection by facilitating RIG-I-like receptor signaling*. Cell Rep, 2021. **34**(3): p. 108631.
6. Li, S.Z., et al., *Phosphorylation of MAVS/VISA by Nemo-like kinase (NLK) for degradation regulates the antiviral innate immune response*. Nat Commun, 2019. **10**(1): p. 3233.
7. Yan, Z., et al., *The protein arginine methyltransferase PRMT1 promotes TBK1 activation through asymmetric arginine methylation*. Cell Rep, 2021. **36**(12): p. 109731.
8. Madisen, L., et al., *A robust and high-throughput Cre reporting and characterization system for the whole mouse brain*. Nat Neurosci, 2010. **13**(1): p. 133-40.
9. Wang, X., et al., *TRIM31 facilitates K27-linked polyubiquitination of SYK to regulate antifungal immunity*. Signal Transduct Target Ther, 2021. **6**(1): p. 298.
10. Zhang, H., et al., *MiRNA-543 promotes osteosarcoma cell proliferation and*

- glycolysis by partially suppressing PRMT9 and stabilizing HIF-1alpha protein.* Oncotarget, 2017. **8**(2): p. 2342-2355.
0. Hadjikyriacou, A., et al., *Unique Features of Human Protein Arginine Methyltransferase 9 (PRMT9) and Its Substrate RNA Splicing Factor SF3B2.* J Biol Chem, 2015. **290**(27): p. 16723-43.
 1. Duan, S., et al., *FBXO11 targets BCL6 for degradation and is inactivated in diffuse large B-cell lymphomas.* Nature, 2012. **481**(7379): p. 90-3.
 2. Fielenbach, N., et al., *DRE-1: an evolutionarily conserved F box protein that regulates C. elegans developmental age.* Dev Cell, 2007. **12**(3): p. 443-55.
 3. Bedford, M.T. and S.G. Clarke, *Protein arginine methylation in mammals: who, what, and why.* Mol Cell, 2009. **33**(1): p. 1-13.
 4. Nishioka, K. and D. Reinberg, *Methods and tips for the purification of human histone methyltransferases.* Methods, 2003. **31**(1): p. 49-58.
 5. Jansen, S., et al., *De novo variants in FBXO11 cause a syndromic form of intellectual disability with behavioral problems and dysmorphisms.* Eur J Hum Genet, 2019. **27**(5): p. 738-746.

REVIEWERS' COMMENTS

Reviewer #1 (Remarks to the Author):

The authors have addressed all of my concerns. They now convincingly show the methylation of MAVS by PRMT9 with its functional consequences. It's a nice discovery.

Reviewer #2 (Remarks to the Author):

All my previous concerns have been adequately addressed.

Reviewer #3 (Remarks to the Author):

The authors carefully addressed the questions and suggestions raised by all three reviewers. It is appreciated that also several new in vivo studies were performed. The revised version of the manuscript significantly improved. Reviewer 3 would like to thank the authors for clarifying some ambiguities regarding the designation of FBXO11 and PRMT9. Considering the fact that some authors in the FBXO11 field in a recent study still claimed the synonymous usage of FBXO11 and PRMT9 (Jansen et al., 2019) a clear statement by the authors of this study that FBXO11 was previously erroneously named PRMT9 would certainly help the interested readers who are not so deeply involved in the PRMT9 field.

Title: The protein arginine methyltransferase PRMT9 attenuates MAVS activation through arginine methylation

Point by point responses to reviewers' comments:

Reviewer #1 (Remarks to the Author):

The authors have addressed all of my concerns. They now convincingly show the methylation of MAVS by PRMT9 with its functional consequences. It's a nice discovery.

Answer: We thank the reviewer for the acceptance of our manuscript.

Reviewer #2 (Remarks to the Author):

All my previous concerns have been adequately addressed.

Answer: We are delighted to know that our revised manuscript has addressed the concerns of reviewer #2.

Reviewer #3 (Remarks to the Author):

The authors carefully addressed the questions and suggestions raised by all three reviewers. It is appreciated that also several new in vivo studies were performed. The revised version of the manuscript significantly improved. Reviewer 3 would like to thank the authors for clarifying some ambiguities regarding the designation of FBXO11 and PRMT9. Considering the fact that some authors in the FBXO11 field in a recent study still claimed the synonymous usage of FBXO11 and PRMT9 (Jansen et al., 2019) a clear statement by the authors of this study that FBXO11 was previously erroneously named PRMT9 would certainly help the interested readers who are not so deeply involved in the PRMT9 field.

Answer: We sincerely appreciate the reviewer's valuable advice during the review process.